# Dynamic estimation of the attentional field from visual cortical activity

**Ilona M Bloem[†‡], Leah Bakst[†], Joseph T McGuire, Sam Ling***

Department of Psychological & Brain Sciences, Boston University, Boston, United States

**eLife Assessment**

This **valuable** study addresses a gap in our understanding of how the size of the attentional field is represented within the visual cortex. The evidence supporting the role of visual cortical activity is **convincing**, based on a novel modeling analysis of fMRI data. The results will be of interest to psychologists and cognitive neuroscientists.

**\*For correspondence:**
samling@bu.edu

[†]These authors contributed equally to this work

**Present address:** [‡]Computational Cognitive Neuroscience and Neuroimaging, Netherlands Institute for Neuroscience, Royal Netherlands Academy of Arts and Sciences, Amsterdam, Netherlands

**Competing interest:** The authors declare that no competing interests exist.

**Abstract** Navigating around the world, we must adaptively allocate attention to our surroundings based on anticipated future stimuli and events. This allocation of spatial attention boosts visuocortical representations at attended locations and locally enhances perception. Indeed, spatial attention has often been analogized to a 'spotlight' shining on the item of relevance. Although the neural underpinnings of the locus of this attentional spotlight have been relatively well studied, less is known about the size of the spotlight: to what extent can the attentional field be broadened and narrowed in accordance with behavioral demands? In this study, we developed a paradigm for dynamically estimating the locus and spread of covert spatial attention, inferred from visuocortical activity using fMRI in humans. We measured BOLD activity in response to an annulus while participants (four female, four male) used covert visual attention to determine whether more numbers or letters were present in a cued region of the annulus. Importantly, the width of the cued area was systematically varied, calling for different sizes of the attentional spotlight. The deployment of attention was associated with an increase in BOLD activity in corresponding retinotopic regions of visual areas V1–V3. By modeling the visuocortical attentional modulation, we could reliably recover the cued location, as well as a broadening of the attentional modulation with wider attentional cues. This modeling approach offers a useful window into the dynamics of attention and spatial uncertainty.

## Introduction

We bounce attention around all the time. Take, for instance, when we're monitoring oncoming traffic while driving. It isn't sufficient to attend to the single most likely source of traffic. Instead, attention adaptively broadens and narrows to cover the anticipated spatial distribution of relevant events. The need to spread attention across different swaths of the visual field is driven, to a large degree, by spatial uncertainty: statistical regularities give us a general sense as to where something useful might happen, and this evolves from moment to moment. We navigate this uncertainty by dynamically deploying spatial attention.

Covert spatial attention improves behavioral performance at attended locations at the cost of performance at unattended locations (*Posner, 1980*), leading to a common metaphor that spatial selective attention acts as a 'spotlight' or 'zoom lens' (*Shaw and Shaw, 1977*; *Posner, 1980*; *Eriksen and St. James, 1986*; *Carrasco, 2011*). This attentional 'spotlight' is characterized by a

specific size and location and traverses the visual field based on behavioral demands (*Eriksen and St. James, 1986*; *Castiello and Umiltà, 1990*), selectively boosting information at the attended location within the visual system while suppressing information elsewhere. Animal studies have observed multiplicative increases in visuocortical neural responses at attended locations (*McAdams and Maunsell, 1999*; *Maunsell, 2015*), and human neuroimaging studies have found similar focal modulations of population responses (*Kastner et al., 1998*; *Brefczynski and DeYoe, 1999*; *McMains and Somers, 2004*; *Datta and DeYoe, 2009*; *Sprague and Serences, 2013*; *Puckett and DeYoe, 2015*; *Samaha et al., 2016*; *Shioiri et al., 2016*; *Bloem and Ling, 2019*; *Bartsch et al., 2023*).

While neural modulation at the locus of attention has been relatively well studied, less is known regarding the neural signatures of the size of the attentional field (*Yeshurun, 2019*). Spreading attention over a larger region of visual space can decrease behavioral performance, but only a handful of studies have interrogated associated effects within visual cortex (*Müller et al., 2003*; *Herrmann et al., 2010*; *Itthipuripat et al., 2014*; *Feldmann-Wüstefeld and Awh, 2020*). This is surprising, as the spatial distribution of the attentional field is a key feature in an influential theoretical model of attention (*Reynolds and Heeger, 2009*). The model assumes that the size of the attentional field can be adjusted based on task demands and that the interaction between attentional field size and stimulus-related factors can predict observed attentional gain effects.

While the studies that have experimentally manipulated the attentional field size found evidence congruent with this prominent theory (*Herrmann et al., 2010*; *Itthipuripat et al., 2014*; *Kınıklıoğlu and Boyaci, 2022*), few studies have directly investigated the spatial extent of the attentional window and its concomitant neural representation. One neuroimaging study revealed that the attentional field expanded in the face of greater task-related uncertainty (*Herrmann et al., 2010*), while other studies showed that the responsive area of visual cortex increased in size, coupled with a decrease of the overall population response (*Müller et al., 2003*; *Feldmann-Wüstefeld and Awh, 2020*). While these studies are consistent with the notion that the attentional field size can be detected in visual cortex, methods for dynamically recovering location and field size from moment to moment are lacking.

In this study, we developed a paradigm that allowed us to dynamically characterize the spatial tuning of spatial attention across the visual field. Using fMRI in humans, we examined whether attentional modulation of the BOLD response spanned a larger area of visual cortex when participants were cued to attend to a larger region of space. Behavioral performance confirmed that participants could successfully allocate their attention to different-sized swaths of the visual field. This deployment of attention was associated with a modulation in cortical activity in the corresponding retinotopic areas of visual cortex. By modeling the location and spread of the visuocortical modulation, we dynamically recovered the cued location from the attentional activity with a high degree of fidelity, together with a broadening of the attentional modulation for wider attentional cues.

## Results

### Behavioral performance indicates effective deployment of covert spatial attention

We set out to investigate the spatial distribution of attentional modulation within visual cortex. To do so, we first ensured that participants (*n*=8) could successfully allocate covert spatial attention to cued portions of the visual field. During the experiment, participants' task was to fixate the center of the screen and report whether there were more *numbers* or *letters* in a cued peripheral region (*Figure 1a*). The cued region varied in location and width: it could be centered on any of 20 polar angles and could span any of four widths (18°, 54°, 90°, and 162° of polar angle). Task performance indicated that participants used the cue effectively, as the proportion of correct responses was significantly above chance for all width conditions (*Figure 1b*; t-test, all p<0.001). We verified, with eye tracking, that participants performed the task using peripheral vision while maintaining central fixation. The upper bound of the 95% CI for each participant's average gaze eccentricity ranged from 0.29° (degrees of visual angle) to 0.64° (mean = 0.48°; *Figure 1c*), suggesting that gaze did not exceed the cue annulus at fixation and that participants used covert spatial attention to perform the task.

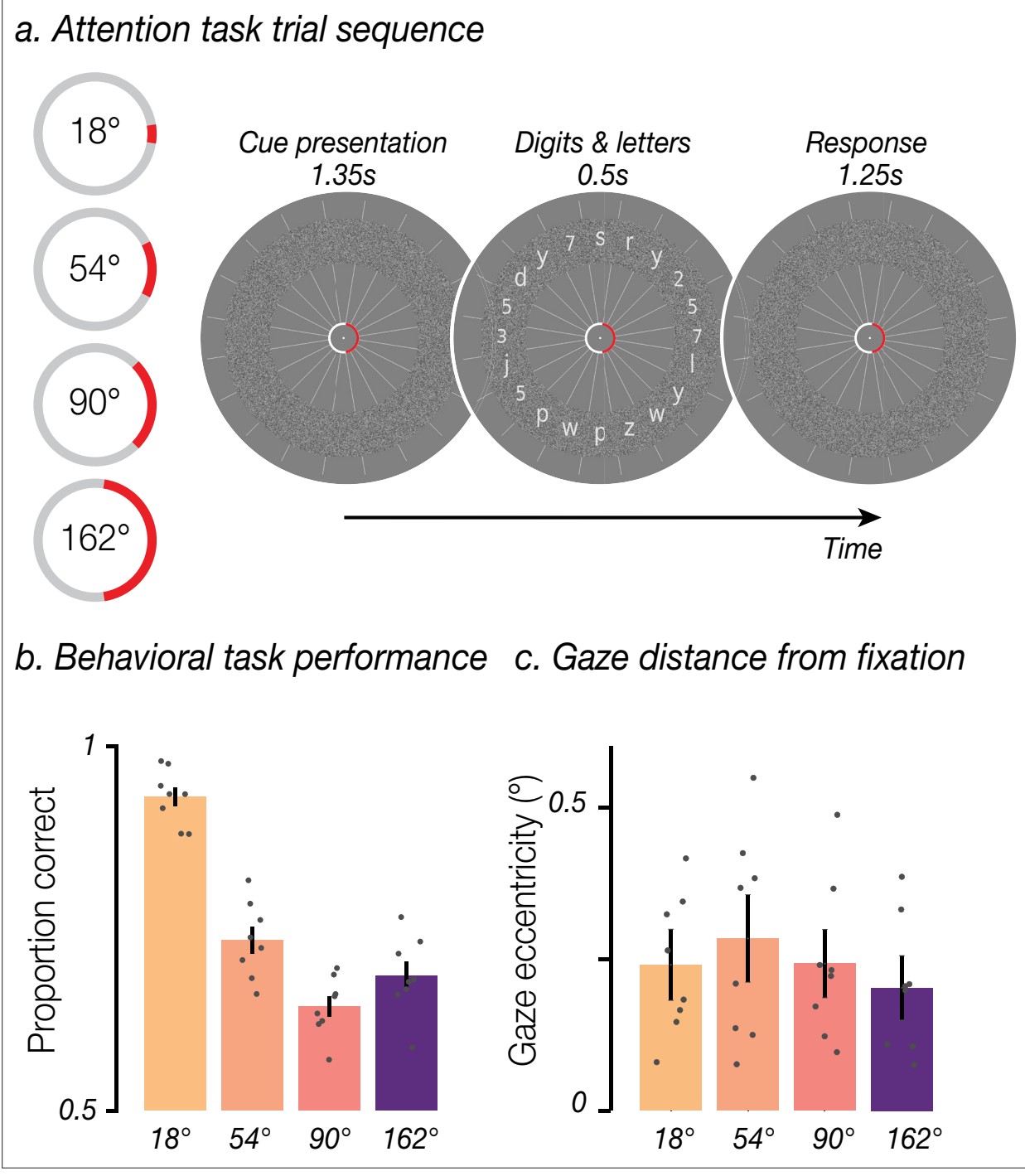

**Figure 1.** Experimental design and behavioral performance. (**a**) Task schematic. Participants were instructed to maintain central fixation and use covert spatial attention to determine whether there were more numbers or letters present within a cued region of a white noise annulus. On each trial, the red cue was displayed alone for 1.35 s and remained present throughout the trial. Twenty digits and letters were then presented for 0.5 s, equally spaced and overlaid on the annulus. Participants had 1.25 s to indicate via button press whether more digits or letters were present in the cued region. The cue remained stable for five trials (10 TRs, 15.5 s), had a width of 1, 3, 5, or 9 segments (18°, 54°, 90°, or 162°), and was centered on any of the 20 digit/letter slots. (**b**) Behavioral task performance: group mean accuracy for each cue width. Error bars are SEM; gray circles show individual participants (n=8). (**c**) Group mean gaze eccentricity (in degrees of visual angle) for each cue width, conventions as in (**b**).

## Attentional modulation of BOLD responses broadens with cue width

We assessed the spatial distribution of attention by visualizing how the BOLD response was modulated by the location and width of the cue. To do so, we used each voxel's population receptive field (pRF) to project BOLD responses for each attentional cue into the visual field. The resulting 2D visual field maps were averaged across trials for each cue width by rotating the maps, so the attentional cue aligned to 0° polar angle (right horizontal meridian). The reconstructed visual field maps revealed that increasing cue width led to a concomitant broadening of attentional modulation in cortex (*Figure 2a*). While this pattern was evident in all three early visual regions (V1–V3), the effect appeared to strengthen when ascending the visuocortical hierarchy.

Next, we computed the one-dimensional (1D) profile of attentional modulation at a fixed eccentricity. We were able to do this because we manipulated the location of the attentional field only as a function of polar angle, so all cues directed the attentional field to iso-eccentric locations. We selected voxels with pRFs that overlapped the white noise annulus and sorted them according to their polar angle preference.

For visualization purposes, the spatial response modulations were recentered to align all cues at 0° polar angle and averaged across trials for each cue width separately. Much like in the visual field reconstructions, there was a clear attentional modulation centered on 0°, which broadened and shifted downward with cue width – a pattern that was particularly evident in area V3 (*Figure 2b*).

## Dynamic model-based recovery of the attentional field

We next applied a modeling approach to estimate the location and width of attentional modulation, allowing us to further investigate the spread of attention in visual cortex. To do this, we averaged the spatial response profiles across TRs within each 10-TR block, in which the cue maintained a consistent location and width, yielding between 27 and 53 averaged spatial response profiles per participant for each width condition. We fit a generalized Gaussian function to each of these spatial profiles to estimate the location and width of attentional modulation per spatial profile (see *Figure 3a*). The width of attentional modulation was quantified in terms of the full width at half maximum (FWHM) of the best fitting model prediction (see *Figure 3b*).

Can we dynamically recover the attentional field from activity within visual cortex? Model fits explained a substantial proportion of variance in the spatial profiles of BOLD activity (**V1**: for 18° cues, mean [standard deviation] of $R^2$=0.42 [0.03]; for 54° cues, 0.43 [0.03]; for 90° cues, 0.44 [0.03]; for 162° cues, 0.42 [0.03]; **V2**: for 18° cues, 0.51 [0.05]; for 54° cues, 0.54 [0.05]; for 90° cues, 0.54 [0.04]; for 162° cues, 0.55 [0.04]; **V3**: for 18° cues, 0.50 [0.03]; for 54° cues, 0.56 [0.04]; for 90° cues, 0.55 [0.03]; for 162° cues, 0.51 [0.02]). To interpret the estimated model parameters, we excluded the bottom 20% of fits based on a pooled $R^2$ across V1, V2, and V3, leaving roughly equal proportions of included blocks across cue width conditions (18°: mean [standard deviation]=0.78 [0.04], 54°: 0.83 [0.05], 90°: 0.83 [0.04], 162°: 0.77 [0.07]).

To assess how well the model-estimated attentional field matched the cued location, we first calculated the angular error between the cue center and the model's estimated location parameter. The angular error distribution across blocks, separated by width condition, is shown in *Figure 4* for one example participant to display block-to-block variation. The model reliably captured the location of the attentional field with low angular error and with no systematic directional bias. This result was observed across participants. We next examined the absolute angular error to assess the overall accuracy of our estimates. The group mean absolute angular error in V1 was 41.9° (SEM = 2.86°), in V2 was 32.2° (2.31°), and in V3 was 24.7° (1.54°). Additionally, the absolute angular error did not vary linearly with the width of the cue in V1 or V2 (regression slopes tested against zero at the group level using a t-test; V1: $t(7)$=0.65, p=0.537; V2: $t(7)$=1.24, p=0.253; *Figure 5*). In V3, we observed a small but statistically significant increase in absolute angular error associated with greater cue widths (mean slope = 1.4, $t(7)$=4.18, p=0.004).

Next, we evaluated the width of the attentional field by visualizing the distribution of FWHM for the same example participant (*Figure 4*) and at the group level (*Figure 5*). Confirming the broadening of the attentional field observed in the visual field reconstruction maps, we found that the estimated FWHM increased with greater cue widths in V2 and V3 (V2 $t(7)$=5.63, p<0.001; V3 $t(7)$=6.49, p<0.001). The effect was not statistically significant in V1 ($t(7)$=1.68, p=0.136).

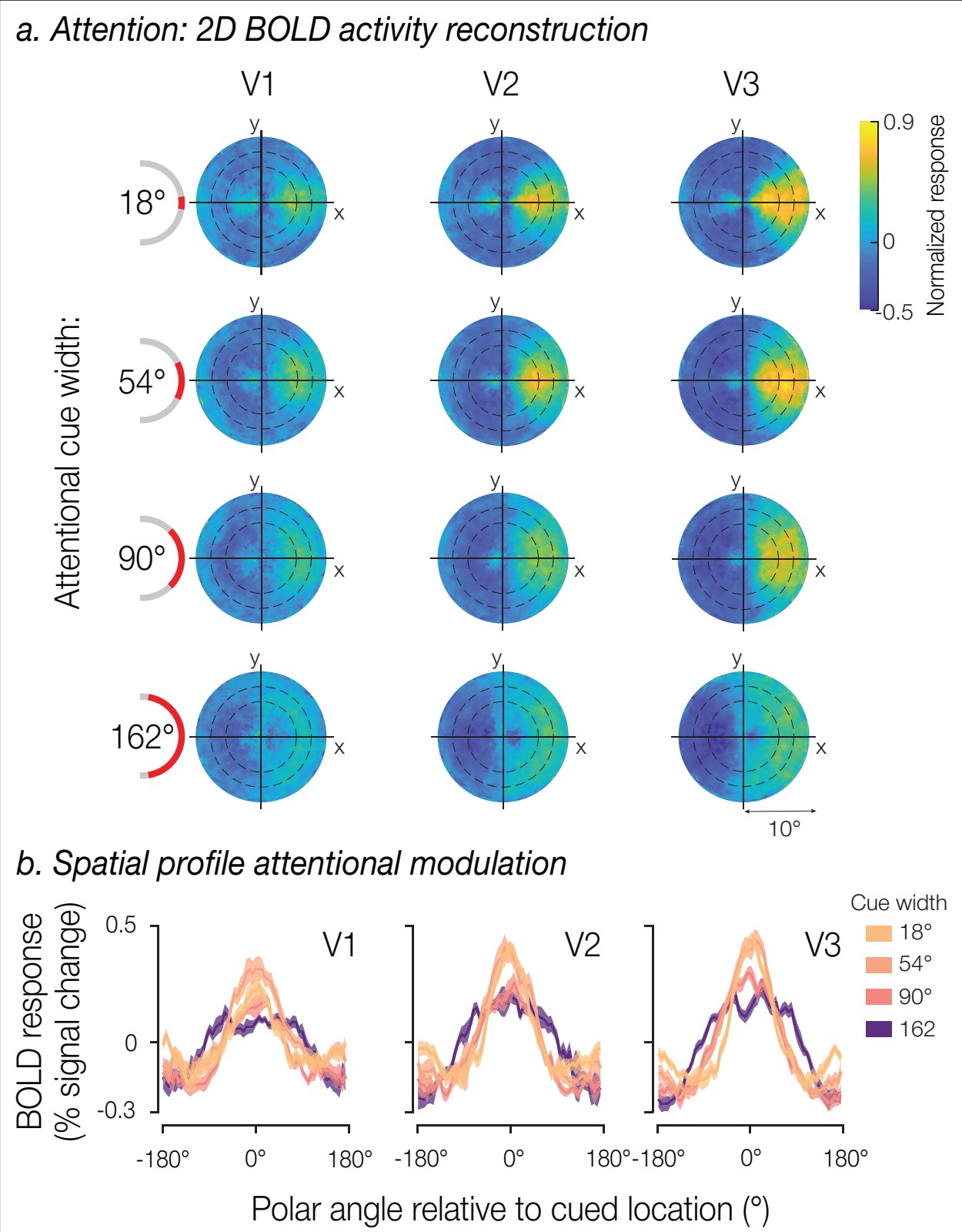

**Figure 2.** Spatial distribution of attentional modulation. (**a**) BOLD response projected into the visual field for each attentional cue width. Heatmaps represent the group mean BOLD activity using each voxel's population receptive field (pRF) location within the visual field, shown separately for V1, V2, and V3. Maps were rotated to align all attentional cue locations to 0° polar angle (rightward). Concentric circles indicated by black dashed lines represent the location of the white noise annulus. (**b**) Average spatial modulation profiles at the eccentricity of the annulus. The spatial profiles were

*Figure 2 continued on next page*

*Figure 2 continued*

recentered to 0° polar angle based on the cue location. Solid lines represent the group mean BOLD response and shaded regions the SEM across participants (n=8).

We also assessed the gain of the attentional modulation in the model (*Figures 4 and 5* for the example participant and group data, respectively). We observed no significant relationship between gain and cue width in V1 and V2 (V1 $t(7)=-0.54$, p=0.605; V2 $t(7)=-2.19$, p=0.065), though we did find a significant effect in V3, illustrating that gain decreases with cue width ($t(7)=-3.12$, p=0.017). We also found that the overall gain was greater in V2 and V3 compared to V1 (paired t-test, both p≤0.01).

Finally, we examined the baseline offset (*Figure 4* example participant, and *Figure 5* group data). No significant relationship was observed between cue width and baseline offset in any of the three brain regions (V1, $t(7)=-1.05$, p=0.330; V2, $t(7)=-2.00$, p=0.086; V3, $t(7)=-1.61$, p=0.152).

## Temporal interval analysis

In the previous analyses, we leveraged the fact that the attentional cue remained constant for five-trial blocks (spatial profiles were computed by averaging BOLD measurements across a block of 10 TRs). We next examined the degree to which we were able to recover the attentional field on a moment-by-moment (TR-by-TR) basis. To do this, we systematically adjusted the number of TRs that contributed to the averaged spatial response profile. To maintain a constant number of observations across the temporal interval conditions, we randomly sampled a subset of TRs from each block. This allowed us to determine the amount of data needed to recover the attentional field, with a goal of examining the usability of our modeling approach in future paradigms involving more dynamic deployment of spatial attention.

When we systematically varied the number of TRs included for each model fit (1, 2, 3, 5, or 10 TRs), we found a significant effect of cue width on recovered FWHM when averaging two or more TRs in V3 (all $t(7)≥2.38$, all p≤0.049), and 10 TRs in V2 (results as reported in prior section; *Figure 6a*). As described above, V1 did not reliably show a significant relationship between cue width and FWHM, even when averaging 10 TRs. We found that increasing the number of TRs had a small but significant

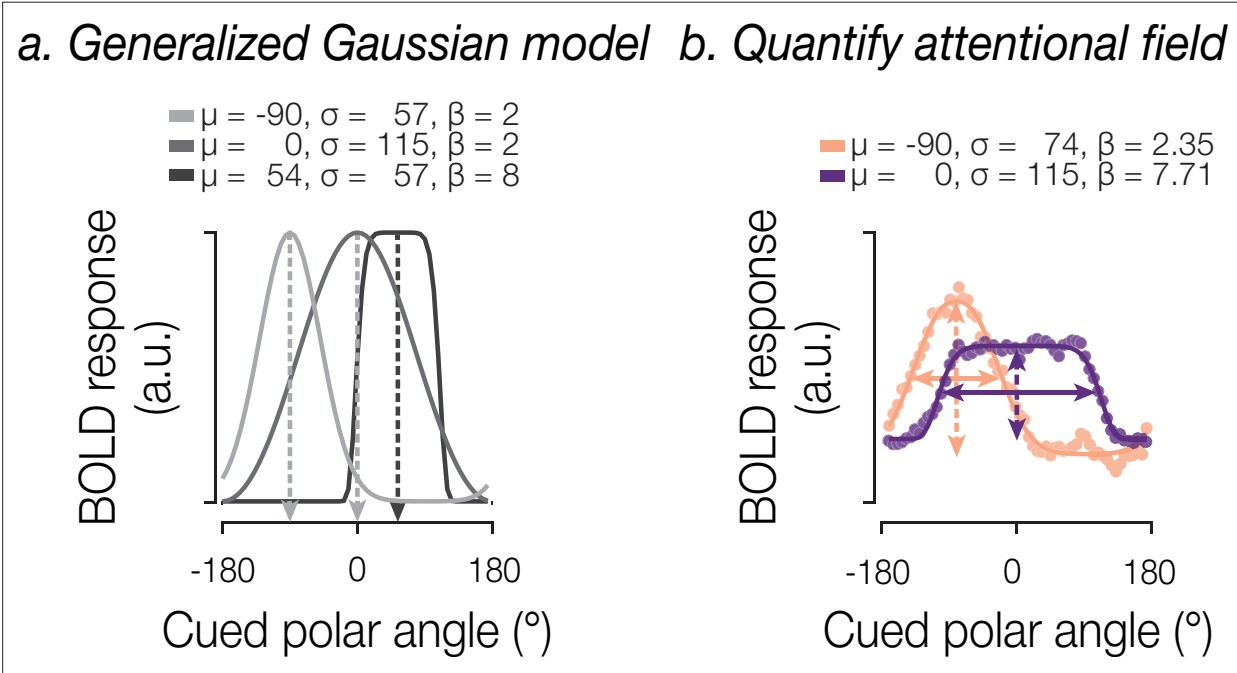

**Figure 3.** Modeling approach. (**a**) The generalized Gaussian model is characterized by parameters for location ($\mu$), scale ($\sigma$), and shape ($\beta$). (**b**) Example model fits for two spatial profiles. Dots indicate BOLD response for two attentional cues differing in position and width. Solid lines indicate the best fitting model estimate. To quantify the attentional field, we extracted the location and gain (dashed arrows), as well as the width (full width at half maximum (FWHM); solid arrows).

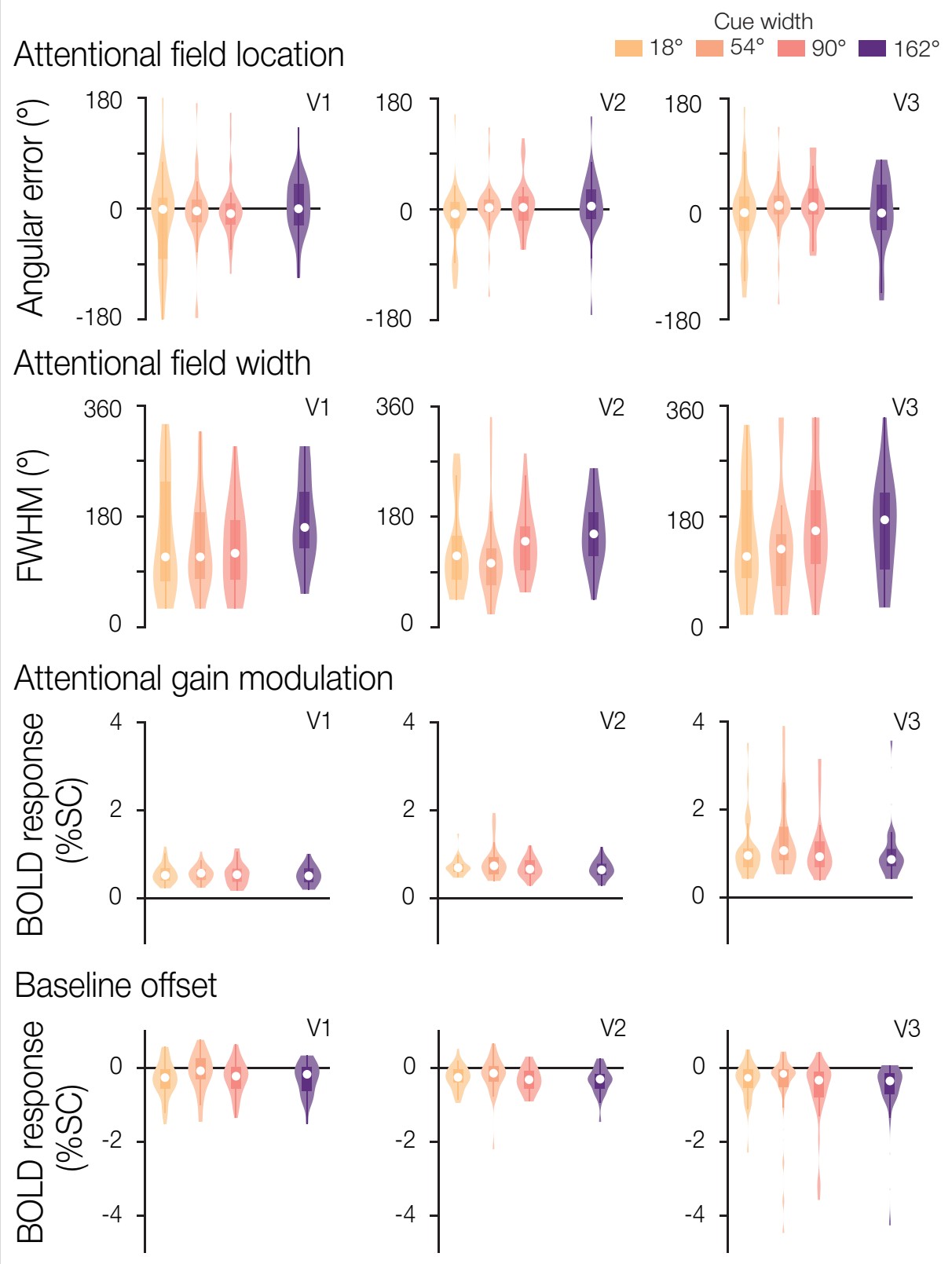

**Figure 4.** Attentional field parameter estimates for an example participant. The full parameter estimate distributions across blocks for location, width, gain, and baseline are shown for one example participant in V1, V2, and V3. Median parameter estimates are shown by the white points, with the box plot representing the 25th to 75th percentile, and whiskers extending to all non-outlier points.

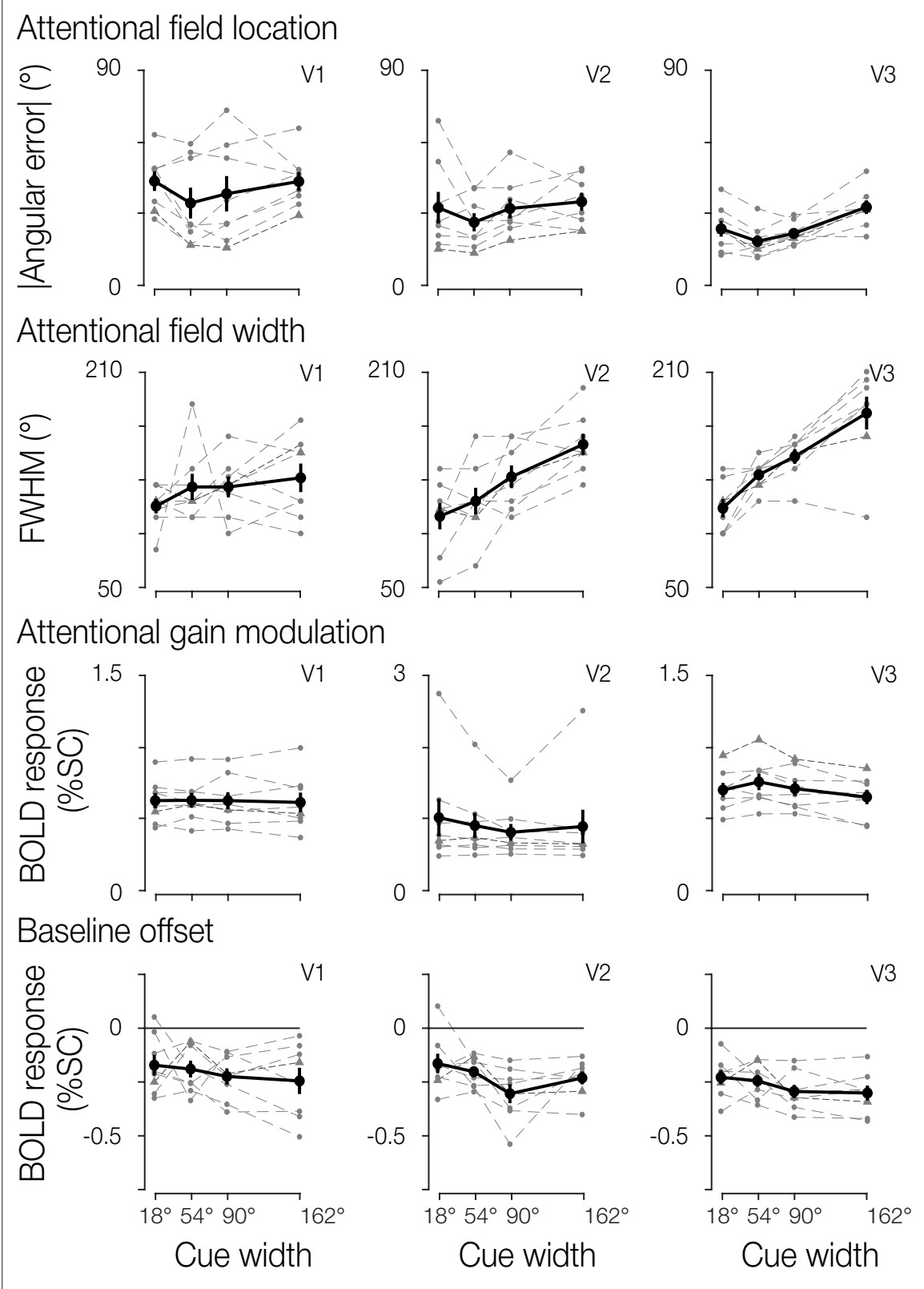

**Figure 5.** Attentional field parameter estimates. Group results for location, width, gain, and baseline estimates. Overall group mean and SEM (n=8) are shown in solid black, separated by cue width and brain region. Individual participant median estimates are shown in gray. The example participant from *Figure 4* is indicated by a denser dashed dark gray line with triangle symbols to aid in comparison.

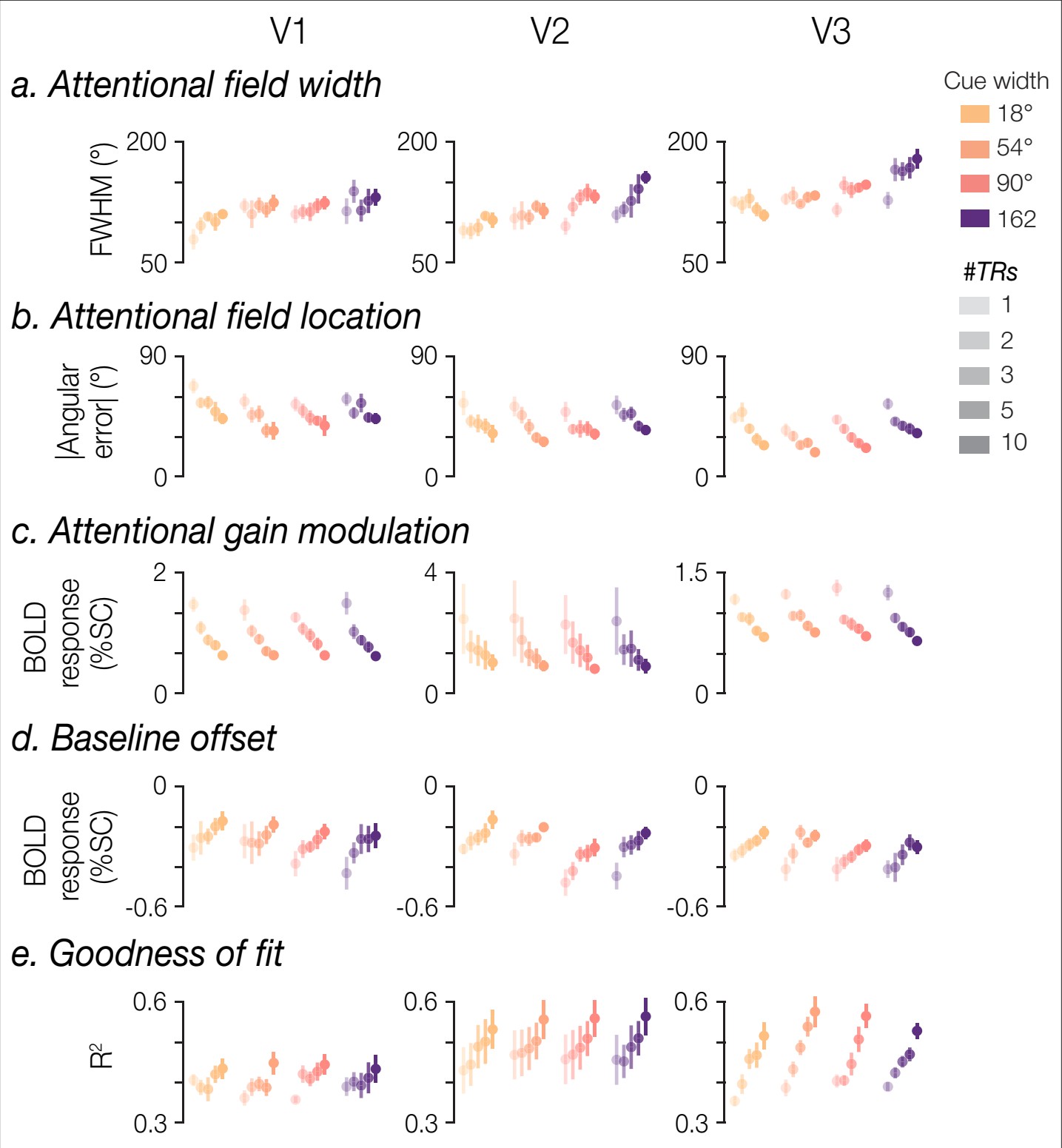

**Figure 6.** Effect of number of TRs. Model fits were computed using BOLD data averaged across different temporal intervals (1, 2, 3, 5, or 10 TRs). Group means (with SEM (n=8)) are plotted for (**a**) full width at half maximum (FWHM), (**b**) absolute angular error, (**c**) gain, (**d**) baseline offset, and (**e**) $R^2$, separated by cue width, brain region, and the number of TRs used for each model fit.

positive effect on FWHM estimates in V2 and V3 (V2, mean slope = 2.7, $t(7)$=2.95, p=0.021; V3, mean slope = 1.16, $t(7)$=3.22, p=0.015), although a significant effect was not observed in V1 ($t(7)$=1.82, p=0.111).

The number of TRs significantly affected the absolute angular error associated with the estimated location of the attentional field (*Figure 6b*). Error magnitude decreased with TRs in all three visual regions (all $t(7)$≤–4.48, all p≤0.003), suggesting that more data yielded more accurate estimates, though absolute angular error remained consistently below chance (90°) even when fitting the model to single-TR BOLD responses. Absolute angular error remained stable across width conditions in V1 and V2 (V1, $t(7)$=–0.55, p=0.598; V2, $t(7)$=1.92, p=0.098), though we found that larger cue width had a small but significant effect associated with larger errors in V3 (mean slope = 0 .02, $t(7)$=3.28, p=0.014).

The estimated gain of the attentional modulation showed a dependence on number of TRs, with more TRs associated with lower gain estimates in V1 and V3 (V1, $t(7)$=–7.21, p<0.001; V3, $t(7)$=–9.97, p<0.001), though this was not clearly observed in V2 ($t(7)$=–1.60, p=0.154). There was no evident dependence of gain on cue width in V1 and V2 (V1 $t(7)$=–0.19, p=0.856; V2 $t(7)$=–2.34, p=0.052), though we did observe a significant relationship in V3 ($t(7)$=–2.86, p=0.024; *Figure 6c*).

The baseline offset tended to increase with number of TRs across all three brain regions (V1, $t(7)$=8.79, p<0.001; V2, $t(7)$=6.5, p<0.001; V3, $t(7)$=5.59, p=0.001; *Figure 6d*). Baseline offset did not show a significant dependence on cue width in any region (V1, $t(7)$=1.47, p=0.186; V2, $t(7)$=–2.16, p=0.068; V3, $t(7)$=–1.67, p=0.139).

Finally, the model's goodness of fit improved with more data, with larger $R^2$ associated with greater numbers of TRs included in the average profiles (all $t(7)$≥2.99, all p≤0.020), though all $R^2$ were above 0.3 across all visual regions even for single-TR model fits. We did not observe a dependence of $R^2$ on cue width (all $t(7)$≤1.26, all p≥0.249; *Figure 6e*).

## Width of the attentional field mimics perceptual modulation

While the attentional field broadened as expected when participants were cued to attend to a larger portion of the white noise annulus, the size of the estimated attentional modulation was greater than the true size of the cued region. The cue width varied between 18° and 162°, whereas the width estimate derived from spatial profiles of BOLD modulation varied between 103° and 179° (*Figure 5*). We wondered what the underlying cause of this disparity might be. One possibility is that the BOLD-derived FWHM might tend to overestimate the retinotopic extent of the modulation, perhaps driven by binning and smoothing processing steps to create the 1D spatial profiles. If this were the case, we would expect to obtain similar FWHM estimates when modeling the perceptual modulations as well. Alternatively, the true subjective attentional field might be consistently broader than cued, despite the presence of nearby distractors. If this were the case, modulation driven by perceptual differences should *not* result in the same large FWHM estimates.

To address this, we compared our estimates of the attentional field with equivalent estimates for spatial profiles induced by a perceptual manipulation. In this additional experiment, we varied the contrast intensity of sections of the white noise annulus. Participants were not asked to deploy spatial attention to the stimulus and were instead instructed to perform a color change detection task at fixation. The regions of increased noise contrast matched the attentional cue widths (18°, 54°, 90°, and 162°, plus an additional intermediate width of 126°) and were centered on one of the four cardinal locations (0°, 90°, 180°, 270° polar angle).

As expected, we observed a broadening of the spatial profile of BOLD modulation in all three visual areas as the region of increased contrast widened (*Figure 7a*). Using an identical modeling procedure, we estimated the spatial profile of the *perceptual* BOLD modulation. The model-based estimates revealed that the mean absolute angular error between the model-estimated location and the center of the contrast stimulus had no significant dependence on contrast width in any of the three brain regions (magnitude of all $t(4)$≤0.915, all p≥0.412). The recovered FWHM increased with contrast width in both V1 and V3 (*Figure 7b*; V1, $t(4)$=6.94, p=0.002; V3 $t(4)$=11.34, p<0.001), though this effect was not clearly observed in V2 ($t(4)$=1.37, p=0.242). The estimated gain modulation also did not show a relationship to contrast width in any of the visual areas (magnitude of all $t(4)$≤1.71, all p≥0.163). Finally, we did not observe a significant relationship between contrast width and baseline offset in any visual area (magnitude of all $t(4)$≤1.93, all p≥0.125). In sum, the group results for model estimates revealed that: (1) the model was highly accurate in estimating the location of the contrast increment;

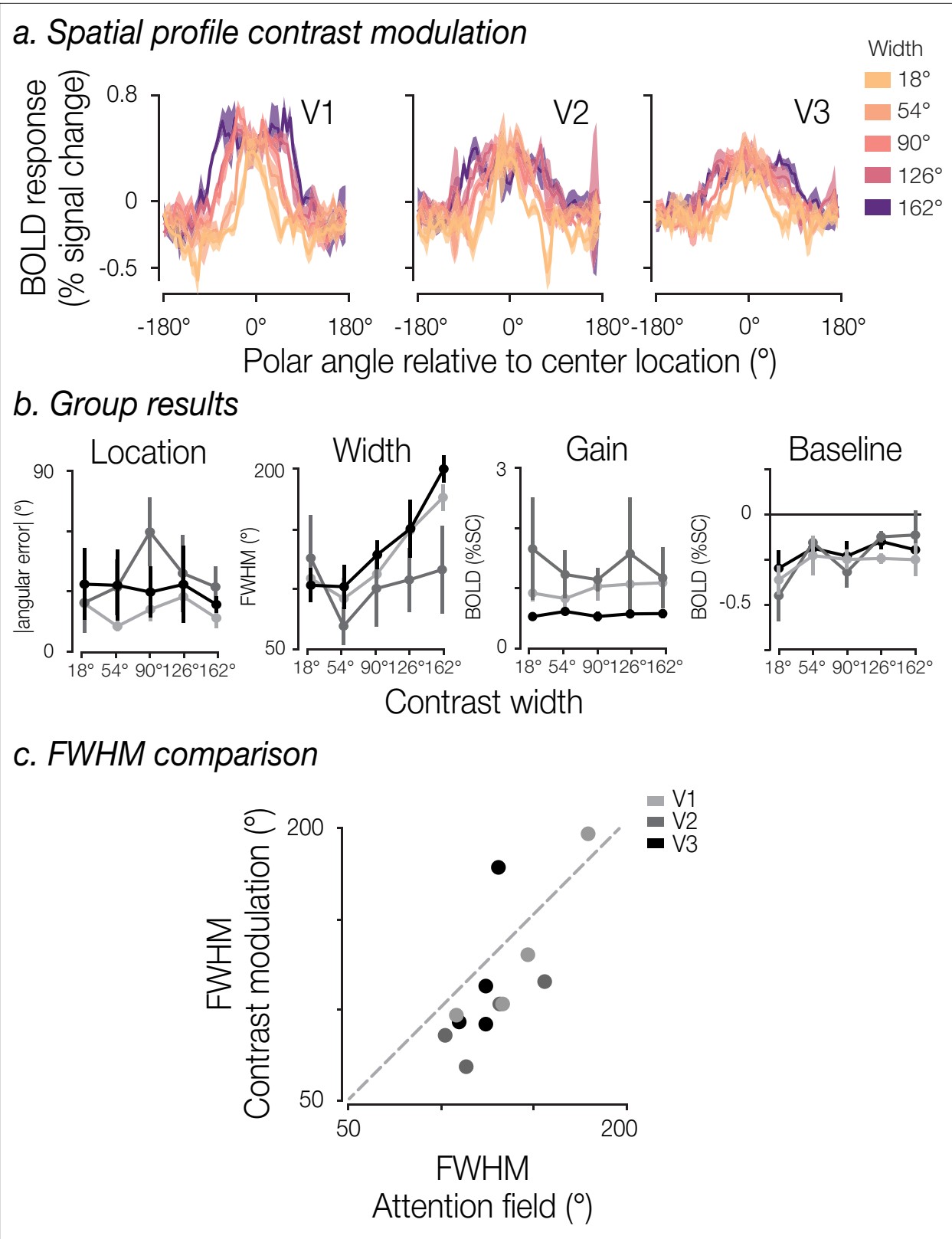

**Figure 7.** Spatial distribution of perceptual modulation. (**a**) Spatial profiles of perceptual modulation. Solid lines represent the group mean BOLD activity and shaded regions the SEM (n=5). (**b**) Group-level parameter estimates. Overall group mean and SEM are shown for the absolute angular error, full width at half maximum (FWHM), gain, and baseline, separated by contrast width and brain region. (**c**) Comparison of FWHM estimates obtained from the attentional manipulation and the physical contrast manipulation. Dot color indicates brain region; each point represents the mean FWHM for a given width condition across participants.

(2) FWHM of the spatial profiles broadened across contrast widths, (3) the gain and baseline remained stable across contrast widths (*Figure 7b*).

Mirroring the results from the attentional manipulation, FWHM estimates systematically exceeded the nominal size of the perceptually modulated region of the visual field. Comparing the estimated FWHMs of the perceptual and attentional spatial profiles (*Figure 7c*) revealed that the estimated widths were highly correlated (Pearson correlation *r*=0.749 across width conditions and visual regions), and though the estimated FWHMs from the perceptual task appear to be smaller, they did not significantly differ from the FWHMs derived from the attentional task (t-test p=0.181). Importantly, the relative differences in FWHM show meaningful effects of both cue and contrast width in a similar manner for attentional and perceptual forms of modulation.

## Discussion

We investigated the topographic spread of spatial attention in human visual cortex by characterizing the spatial profile of BOLD responses while participants attended to different portions of the visual field. Behavioral performance confirmed that participants used the fixation cue to dynamically allocate attention to different swaths of the visual field. Attention allocation was associated with a modulation in the BOLD response in corresponding retinotopic areas of visual cortex. To characterize the topography of that modulation, our approach involved selecting voxels with pRF preferred eccentricities that overlapped our white noise annulus and organizing those voxels into 1D profiles of attentional modulation as a function of preferred polar angle. This allowed us to model the location and spread of the attentional field and test how well it tracked the nominal location and width of the cue presented at fixation. Using a generalized Gaussian model, the cued location could be recovered with high fidelity. We observed a broadening of the estimated attentional field in areas V2 and V3 with the cue width, suggesting our method was capable of dynamically recovering the location and size of the attentional field from moment to moment. We also found that the estimated spatial spread of the attentional modulation (as indicated by the recovered FWHM) was consistently wider than the cued region itself. We therefore compared the spread of the attention field with the spatial profile of a *perceptually* induced width manipulation. The results were comparable in both the attentional and perceptual versions of the task, suggesting that cueing attention to a region results in a similar 1D spatial profile to when the stimulus contrast is simply increased in that region.

This work builds on the concept of an attentional 'spotlight' or 'zoom lens' that has long been theorized to aid in spatial attention (*Shaw and Shaw, 1977*; *Posner, 1980*; *Eriksen and St. James, 1986*; *Carrasco, 2011*). By flexibly adjusting and shifting the focus of the spotlight, visual representations are selectively enhanced in a region of the visual field. However, the empirical evidence demonstrating that attention can change its *spread* across the visual field by modulating brain responses is surprisingly lacking (*Yeshurun, 2019*). Our understanding of how the attentional window interacts with spatial representations is mainly based on behavioral reports (*Gobell et al., 2004*; *Palmer and Moore, 2009*; *Herrmann et al., 2010*; *van Beilen et al., 2011*; *Taylor et al., 2015*; *Huang et al., 2017*; *Kınıklıoğlu and Boyaci, 2022*), but see *Müller et al., 2003*; *Hopf et al., 2006*; *Itthipuripat et al., 2014*; *Tkacz-Domb and Yeshurun, 2018*; *Feldmann-Wüstefeld and Awh, 2020*. We introduced a novel modeling approach that recovered the location and the size of the attentional field. Our data show that the estimated spatial spread of attentional modulation (as indicated by the recovered FWHM) consistently broadened with the cue width, replicating prior work (*Müller et al., 2003*; *Herrmann et al., 2010*). Our results go beyond prior work by linking the spatial profiles to pRF estimates, allowing us to quantify the spread of both attentional and perceptual modulation in degrees of polar angle. Interestingly, the FWHM estimates for the attentional and perceptual spatial profiles were highly similar. Additionally, for area V3, we replicate that the population response magnitude decreased with cue width (*Müller et al., 2003*; *Feldmann-Wüstefeld and Awh, 2020*). One innovation of our method is that it directly reconstructs attention-driven modulations of responses in visual cortex, setting it apart from other methods, such as inverted encoding models (e.g. *Sprague and Serences, 2013*). Finally, we demonstrated that our method has potential to be used in more dynamic settings, in which changes in the attentional field need to be tracked on a shorter timescale.

The ability to change the size of the attentional field is a crucial component in an influential theoretical model of attention. This model proposes that the interaction between stimulus properties (such as its size and specific features) and the attentional field can explain a wide variety of attentional effects

reported in behavioral and neurophysiological studies (*Herrmann et al., 2010*; *Itthipuripat et al., 2014*; *Bloem and Ling, 2019*; *Jigo et al., 2021*). The present study sought to address this gap, with our results showing that the visuocortical attentional field broadened as we increased the cue width (*Figure 5*). This provides compelling evidence that the attention-related cortical response can, in fact, flexibly vary in its position and spatial distribution.

The observed effects of attentional field width were unlikely to be directly attributable to variation in task difficulty. Participants' task in our study was to discriminate whether more numbers or more letters were presented within a cued region of an iso-eccentric annulus of white noise. For our different cue widths, the ratios of numbers and letters were selected to be as similar as possible given the size and spacing of our stimuli. Changes in accuracy across the three larger cue widths were small and non-monotonic, implying task difficulty was dissociable from width per se. This dissociation bolsters the interpretability of our model fits; nevertheless, future work should further investigate how task difficulty interacts with the spread of the attentional field and the amplitude of attention-related BOLD effects (cf. *Ress et al., 2000*).

In this study, we modeled the attentional field using a 1D distribution. This approach aligned with our experimental design, as the attentional cue was manipulated only as a function of polar angle. However, we know that spatial processing varies substantially as a function of eccentricity. Spatial resolution is highest at the fovea and rapidly drops in the periphery (*Anton-Erxleben and Carrasco, 2013*). The spatial distribution of attention will presumably also vary with eccentricity and will likely take on different functional properties close to the fovea, where spatial resolution is high, compared to the far periphery where spatial resolution is low (*Intriligator and Cavanagh, 2001*; *Jigo et al., 2021*). Future work can help provide a better understanding of the contribution of spatial attention by considering how the attentional field interacts with these well-described spatial variations across the visual field. Measuring the full spatial distribution of the attentional field (across both eccentricity and polar angle) will shed light on how spatial attention guides perception by interacting with the nonuniformity of spatial representations.

The spread of the attentional field likely influences the degree to which spatial resolution at the attended location is transformed, leading to enhanced behavioral performance. Spatial attention was vital for this task, as enhanced spatial perception allowed the participants to better discriminate all stimuli within the cued region (*Anton-Erxleben and Carrasco, 2013*). Future work could unpack the degree to which the size of the attentional field influences the spatial resolution of visual cortical representations (*Klein et al., 2014*; *Vo et al., 2017*; *Tünçok et al., 2024*), and how this influences spatial perception.

Beyond addressing core questions related to the function of spatial attention, this method also lays groundwork for addressing questions about spatial predictive uncertainty and belief updating. Prior work on these topics has relied almost entirely on inferring participants' predictions from their behavior, often requiring participants to report overt point predictions (*Nassar et al., 2010*; *McGuire et al., 2014*; *d'Acremont and Bossaerts, 2016*; *Nassar et al., 2019*), or inferring participants' predictions from their sequences of decisions (*Daw et al., 2006*; *Behrens et al., 2007*; *Payzan-LeNestour and Bossaerts, 2011*; *Payzan-LeNestour et al., 2013*). These approaches have shed light on how we dynamically adapt our learning and belief updating processes over time in differently structured contexts. However, methods for recovering information about full predictive belief distributions have been limited, relying on indirect measurements such as eye movements (*O'Reilly et al., 2013*; *Bakst and McGuire, 2021*; *Bakst and McGuire, 2023*), and physiological measures of uncertainty and surprise in EEG and pupillometry (*Preuschoff et al., 2011*; *Nassar et al., 2012*; *Nassar et al., 2019*). The methods developed here offer a potential way to recover the location and width of a spatial predictive distribution via the attentional field in contexts in which it is unknown a priori and might be dependent on how a given participant has integrated previous sequential evidence. Future work could extend this method to more directly interrogate how predictive uncertainty is represented throughout the brain on a moment-by-moment basis.

In summary, we found evidence that people could dynamically adapt the spread of spatial attention, and that the retinotopic extent of attentional modulation of the BOLD response reflected this dynamic adaptation. These findings address a gap in our understanding of spatial attentional control, supporting core theoretical models of attention. Our modeling approach also lays the groundwork to address further questions related to how the attentional field interacts with the

nonuniformity of spatial representations and how uncertainty in spatial contexts is represented in the human brain.

## Materials and methods

### Participants

Eight healthy adults (four female, four male, mean age = 30) participated in the main attention experiment, five of whom also participated in a second experiment featuring a contrast manipulation. All participants had normal or corrected-to-normal vision. All procedures were approved by the Boston University Institutional Review Board, and informed consent was obtained from all participants.

### Apparatus and stimuli

Participants were presented with stimuli generated using PsychoPy (v1.85.1; *Peirce, 2007*) on a MacBook Pro. The visual stimuli were displayed on a rear-projection screen (subtending ~20°×16° visual angle) using a VPixx Technologies PROPixx DLP LED projector (maximum luminance 306 cd/m$^2$). Participants viewed the screen through a front surface mirror. Participants were placed comfortably in the scanner with padding to minimize head motion.

### Procedure

#### Attentional width manipulation

Participants were instructed to fixate a central point (radius 0.08° visual angle) while dynamic pixel-wise white noise (flickering at 10 Hz, 50% contrast) was presented in the periphery (annulus spanning 4.6° to 7.4° visual angle). The annulus was segmented into 20 bins (18° polar angle per bin) by white grid lines radiating from a white circle at the center of the screen (radius 0.25°), passing behind the annulus, and terminating at 8.5° eccentricity. In the middle of each bin, a number or letter (height: 2.1°) was superimposed on the white noise annulus (see *Figure 1a*). For a subset of the participants (three out of eight), the screen distance inside the scanner was changed; therefore, for those participants, the letter size was 1.86° visual angle, and the white noise annulus spanned 4.1° to 6.5° visual angle. The set of possible letters included all lowercase letters of the Latin alphabet except a, b, e, g, i, o, and u. The set of numbers included 2, 3, 4, 5, 7, and 8.

Participants were cued to attend covertly to a contiguous subset of the bins, and their task was to report, via button press, whether there were more *numbers* or *letters* present within the cued region. The cue was a bold red segment on the central white circle, which corresponded to 1, 3, 5, or 9 bins (18°, 54°, 90°, or 162° polar angle; see *Figure 1a*). The true proportion of letters versus numbers was controlled within each cue width condition. For cued regions of 1 bin, there was either a single number or letter in the bin. For cued regions of 3 bins, the ratio was always 2:1 (either two numbers and one letter or vice versa). For cued regions of 5 bins, the ratio was 3:2, and for cued regions of 9 bins, the ratio was 6:3. The ratios were selected to be as similar as possible given the size and spacing of our stimuli (aside from the one-bin cue, the proportions for the other cues were 0.67, 0.60, and 0.67). Cues could be centered on any of the 20 bins.

Participants completed 8–12 runs of the task (mean = 10.4), with each run lasting 341 s and containing 100 trials. Each cue remained constant for a block of five trials (lasting 15.5 s, 10 TRs), although the letters and numbers within the cued region changed on every trial. Thus, each participant saw 20 unique cues (combinations of cue location and width) per run. Each run began and ended with 15.5 s of the dynamic noise annulus.

During each trial, the cue and white noise annulus were presented alone for 1.35 s. The numbers and letters were then displayed for 0.5 s. Thereafter, the cue and white noise remained visible while the participant had 1.25 s to indicate whether there had been more digits or letters within the cued region, resulting in a total trial duration of 3.1 s (2 TRs). No accuracy feedback was provided during the main experiment. However, all participants completed three training runs with trial-by-trial feedback prior to the scan session. During training runs, the response window was shortened to 1 s and the remaining 0.25 s presented feedback in the form of a change in color of the fixation point (blue for correct responses and orange for incorrect responses).

## Physical contrast manipulation

A subset of participants (*n*=5) also participated in an experiment that enhanced the physical contrast intensity of the dynamic visual noise in segments of the annulus. This additional experiment was carried out during the same scan session and allowed for benchmarking the detectability of stimulus-evoked modulation in visual cortex using our analyses. The stimuli and trial structure were similar to the attentional manipulation. The task differed in the following ways: (1) the contrast of the white noise annulus was increased to 100% for segments of the annulus corresponding to 1, 3, 5, 7, or 9 bins (18°, 54°, 90°, 126°, or 162° polar angle), with a Gaussian rolloff ($\sigma$=15°) that spanned 25% of the furthest included bins and 25% of the adjacent excluded bins; (2) the enhanced segments were always centered on the cardinal directions (0°, 90°, 180°, and 270° polar angle); (3) the contrast increase remained constant for 15.5 s (10 TRs); (4) participants performed a color change detection task at fixation. Each unique combination of four locations and five widths of the contrast enhancement was shown once per run, with the order randomized. To estimate a baseline response, each run started and ended with 15.5 s without contrast modulation. Participants completed two runs total, each lasting 341 s (220 TRs).

Throughout the *physical contrast* runs, participants were instructed to fixate on a central point (radius 0.08° visual angle) and to press a button when the fixation point switched color (alternating white and red). The fixation point remained a color for at least 1 s and then had a 10% probability of switching every 100 ms. No cues were presented for the regions of increased contrast. Additionally, no letters or numbers were superimposed on the white noise annulus.

## pRF mapping

pRF estimates were obtained for each participant in a separate scan session. We used the experimental procedure as described in the Human Connectome Project 7T Retinotopy dataset (*Benson et al., 2018*). Stimuli were composed of a pink noise background with colorful objects and faces at various spatial scales, displayed on a mean luminance gray background. Stimuli were updated at a rate of 15 Hz while participants performed a color change detection task at fixation. Participants viewed two types of mapping stimuli: (1) contracting/expanding rings and rotating wedges; (2) moving bar stimuli (*Dumoulin and Wandell, 2008*; *Kay et al., 2013*). A total of four to six scans (300 TRs) were collected for each participant (two to three scans per stimulus type). In this session, the field of view was restricted to the occipital cortex to maximize signal-to-noise ratio (SNR), thereby limiting the brain regions for which we had pRF estimates to V1, V2, and V3.

## MRI data acquisition

All MRI data were acquired at Boston University's Cognitive Neuroimaging Center (Boston, MA, USA) on a research-dedicated Siemens Prisma 3T scanner using a 64-channel head coil. A scanning session lasted 2 hr.

All functional neuroimaging data were acquired using a simultaneous multislice gradient-echo echo-planar acquisition protocol (*Moeller et al., 2010*; *Setsompop et al., 2012*): 2 mm isotropic voxels; FoV = 212 × 212 mm$^2$; 72 axial slices; TR = 1.55 s; TE = 35.60 ms; flip angle = 72°; multiband acceleration factor 4. We computed distortion field maps by using a spin echo echoplanar protocol with opposite *y*-axis phase encoding directions (2 mm isotropic voxels; FOV = 212 × 212 mm$^2$; TR = 8850 ms; TE = 70.80 ms; flip angle = 90°). During a separate scan session, we acquired a whole-brain anatomical scan using a T1-weighted multi-echo MPRAGE 3d sequence (1 mm isotropic; FoV = 256 × 256 mm$^2$; 176 sagittal slices; TR = 2530 ms; TE = 1.69 ms; flip angle = 7°), and the pRF scans (occipital coverage only; right-left phase encoding; 2 mm isotropic voxels; FoV = 136 × 136 mm$^2$; 36 slices; TR = 1 s; TE = 35.4 ms; flip angle = 64°; multiband acceleration factor 3).

## MRI data analysis

### Structural data preprocessing

Whole-brain T1-weighted anatomical data were analyzed using the standard 'recon-all' pipeline provided by FreeSurfer software (FreeSurfer version 5.3, *Fischl, 2012*), generating cortical surface models, whole-brain segmentation, and cortical parcellations.

## Functional data preprocessing

All analyses were performed in the native space for each participant. First, EPI distortion correction was applied to all fMRI BOLD time-series data using a reverse phase-encode method (*Andersson et al., 2003*) implemented in FSL (*Smith et al., 2004*). All functional data were then preprocessed using FS-FAST (*Fischl, 2012*), including standard motion-correction procedures, Siemens slice timing correction, and boundary-based registration between anatomical and functional volumetric spaces (*Greve and Fischl, 2009*). To facilitate voxel-wise analysis, no volumetric smoothing was performed, and across-run within-modality robust rigid registration was applied (*Reuter et al., 2010*), with the middle time point of the first run serving as the target volume, and the middle time point of each subsequent run used as a movable volume for alignment. Lastly, data were detrended (0.005 Hz high-pass filter) and converted to percent signal change for each voxel independently using custom code written in MATLAB (version 2020b).

## pRF mapping and voxel selection

The time series were analyzed using the analyzePRF toolbox in MATLAB, implementing a compressive spatial summation pRF model (*Kay et al., 2013*). The results of the pRF analysis were used to manually draw boundaries between early visual regions (V1, V2, and V3), which served as our regions of interest (ROIs).

Within each ROI, pRF modeling results were used to constrain voxel selection used in the main experiment. We excluded voxels with a preferred eccentricity outside the bounds of the pRF stimulus (<0.7° and >9.1°), with a pRF size smaller than 0.01°, or with poor spatial selectivity as indicated by the pRF model fit ($R^2$<10%). Following our 2D visualizations (see below), we further constrained voxel selection by only including voxels whose pRF overlapped with the white noise annulus. We included all voxels with an estimated eccentricity within the annulus bounds, as well as voxels with an estimated pRF size that would overlap the annulus.

## 2D visualizations of attentional modulation

To visualize the topography of attentional modulation under different cue widths, we projected the average BOLD responses for a given block (10 TRs with a consistent cue location and width, shifted by 3 TRs [4.65 s] to compensate for the hemodynamic delay) into the visual field using each voxel's pRF location. This method is similar to that described in *Favila et al., 2022*. First, we computed the Cartesian ($x,y$) coordinates from the pRF eccentricity and polar angle estimates for each voxel. Then, within a given ROI, we interpolated the BOLD responses over ($x,y$) space to produce a full-field representation. Each representation was then z-scored to allow for comparison across blocks, cue conditions, and participants. Finally, the representation was rotated so that the center of the cue was aligned to the right horizontal meridian (see *Figure 2a*).

## 1D spatial profile of attentional modulation

We also examined the spatial profile of attentional modulation as a function of polar angle. Voxels with pRFs overlapping the white noise annulus were grouped into 60 bins according to their pRF polar angle estimate (6° polar angle bin width). We computed a median BOLD response within each bin. This facilitated the recentering of each profile to align all cue centers for subsequent combining across trials. To improve the SNR, the resulting profile was smoothed with a moving average filter (width 18° polar angle; see *Figure 2b*).

## Model fitting

We quantified the spatial profile of attentional modulation with a generalized Gaussian model (*Nadarajah, 2005*). The generalized Gaussian function (*G*) combines Gaussian and Laplace distributions:

$$G = exp\left\{ -\left| \frac{x - \mu}{\sigma} \right|^{\beta} \right\} \tag{1}$$

The function has free parameters for location ($\mu$), scale ($\sigma$), and shape ($\beta$). The shape parameter enables the tails of the distribution to become heavier than Gaussian (when $\beta < 2$), or lighter than Gaussian (when $\beta > 2$); as $\beta \to \infty$, the model approaches a uniform distribution.

Next, $G$ was normalized to range between 0 and 1, and vertically scaled and shifted by two additional free parameters for gain ($a$) and baseline offset ($b$):

$$\hat{y} = a \cdot G + b \tag{2}$$

We fit the five free parameters ($\mu, \sigma, \beta, a, b$) using the MATLAB optimization tool *fmincon*, minimizing the squared error between the model prediction and the 1D profile described above. To avoid local minima, we first ran a grid search to find the initialization values with the lowest SSE (six possible values for $\mu$, equally spaced between 0° and 360°, crossed with six possible values for $\sigma$, equally spaced between 9° and 162° polar angle; $\beta = 4$; $a = 1$; $b = 0$). We imposed the following parameter bounds on the search: $\sigma$: [6°, 180° polar angle], $\beta$: [1.8, 50], and $a$: [0, 20]. $\mu$ was unbounded, but was wrapped to remain within [0°, 360°].

From the model fits, we computed the following summary metrics: (1) angular error, defined as the polar-angle distance between the true and estimated location; (2) the FWHM of the best-fitting generalized Gaussian function, which served as our measure of the width of attentional modulation. The FWHM was controlled mainly by the scale parameter ($\sigma$) but also to a lesser degree by the shape parameter ($\beta$; see *Figure 3a*); (3) the gain modulation of the spatial profile ($a$); (4) the baseline offset ($b$); (5) the model's goodness of fit quantified as the percentage of explained variance ($R^2$) in the spatial response profile:

$$R^2 = 1 - \frac{(y - \hat{y})^2}{(y - \bar{y})^2} \tag{3}$$

## Statistical testing

To assess how the attentional cue width manipulation influenced the 1D spatial profile of BOLD modulation, we tested whether the computed summary metrics (absolute angular error, FWHM, gain, and baseline) varied as a function of cue width. Specifically, we performed a linear regression for each metric within each subject and tested whether the slopes differed from zero at the group level using a t-test. This was done independently for each ROI. No multiple comparison correction was applied, as the different tests for each region are treated as separate questions. However, using a threshold of 0.017 for p-values would correct for comparisons across the three brain regions. When testing whether the number of TRs impacted our metrics, the linear regression used both cue width and number of TRs as explanatory variables.

## Eye-position monitoring

Gaze data were collected for all participants using an MR-compatible SR Research EyeLink 1000+ eye tracker sampling at 1 kHz. Data from blink periods were excluded from analysis. Participants maintained fixation throughout the task, with average gaze eccentricity below 0.5° for all participants. Gaze eccentricity did not significantly vary by cued width (pairwise comparison of width conditions using a paired t-test, all p≥0.205 with Bonferroni correction for multiple comparisons) nor location (pairwise comparison, all p≥0.522 with Bonferroni correction for multiple comparisons). Additionally, we examined the number of fixations to the white noise annulus itself. No participant had more than 16 fixations (out of 800–1200 trials) to the annulus during the task, further suggesting that participants successfully maintained fixation.

## Acknowledgements

This work was supported by National Science Foundation grants SMA-1809071, and BCS-1755757, National Institutes of Health grants F32-EY029134, R01-EY028163, and R01-MH126971, Office of Naval Research grant N00014-17-1-2304, and the Center for Systems Neuroscience Postdoctoral Fellowship at Boston University. The equipment used was funded by NSF Major Instrumentation Grant 1625552. The content of this paper does not necessarily represent the official views of the funding agencies. We thank members of the Ling and McGuire labs for providing helpful feedback and comments.

# Additional information

### Funding

| Funder | Grant reference number | Author |
|---|---|---|
| National Science Foundation | SMA-1809071 | Leah Bakst<br>Joseph T McGuire |
| National Science Foundation | BCS-1755757 | Joseph T McGuire<br>Sam Ling |
| National Eye Institute | F32-EY029134 | Leah Bakst<br>Joseph T McGuire |
| National Eye Institute | R01-EY028163 | Sam Ling |
| National Institute of Mental Health | R01-MH126971 | Joseph T McGuire |
| Boston University | Center for Systems Neuroscience Postdoctoral Fellowship | Leah Bakst |

The funders had no role in study design, data collection and interpretation, or the decision to submit the work for publication.

### Author contributions

Ilona M Bloem, Conceptualization, Data curation, Formal analysis, Investigation, Methodology, Software, Visualization, Writing – original draft, Writing – review and editing; Leah Bakst, Conceptualization, Formal analysis, Funding acquisition, Investigation, Methodology, Software, Validation, Visualization, Writing – original draft, Writing – review and editing; Joseph T McGuire, Conceptualization, Methodology, Writing – original draft, Investigation, Funding acquisition, Formal analysis; Sam Ling, Conceptualization, Methodology, Writing – original draft, Investigation, Data curation, Formal analysis

### Author ORCIDs

Ilona M Bloem (iD) https://orcid.org/0000-0002-7926-6500
Leah Bakst (iD) http://orcid.org/0000-0003-2741-5532
Joseph T McGuire (iD) https://orcid.org/0000-0001-6259-0809
Sam Ling (iD) https://orcid.org/0000-0002-6735-2508

### Ethics

Human subjects: All procedures were approved by the Boston University Institutional Review Board, and informed consent was obtained from all participants.

Reviewer #1 (Public review): https://doi.org/10.7554/eLife.104222.3.sa1
Reviewer #2 (Public review): https://doi.org/10.7554/eLife.104222.3.sa2
Author response https://doi.org/10.7554/eLife.104222.3.sa3

---

# Additional files

### Supplementary files

MDAR checklist

### Data availability

All preprocessed fMRI data (extracted time series from each voxel), behavioral data, and eye tracking data reported in this study, as well as all code necessary to reproduce the analysis and recreate the figures are publicly available at Open Science Framework and GitHub (copy archived at *Bloem, 2025*) respectively.

The following dataset was generated:

| Author(s) | Year | Dataset title | Dataset URL | Database and Identifier |
|---|---|---|---|---|
| Bloem I, Bakst L, McGuire JT, Ling S | 2025 | Data and code for: "Dynamic estimation of the attentional field from visual cortical activity" | https://doi.org/10.17605/OSF.IO/VEK9M | Open Science Framework, 10.17605/OSF.IO/VEK9M |

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
