## [Editor Report · eLife Assessment]

This **valuable** study addresses a gap in our understanding of how the size of the attentional field is represented within the visual cortex. The evidence supporting the role of visual cortical activity is **convincing**, based on a novel modeling analysis of fMRI data. The results will be of interest to psychologists and cognitive neuroscientists.

---

## [Referee Report · Reviewer #1 (Public review)]

The authors conducted an fMRI study to investigate the neural effects of sustaining attention to areas of different sizes. Participants were instructed to attend to alphanumeric characters arranged in a circular array. The size of attention field was manipulated in four levels, ranging from small (18 deg) to large (162 deg). They used a model-based method to visualize attentional modulation in early visual cortex V1 to V3, and found spatially congruent modulations of the BOLD response, i.e., as the attended area increased in size, the neural modulation also increased in size in the visual cortex. They suggest that this result is a neural manifestation of the zoom-lens model of attention and that the model-based method can effectively reconstruct the neural modulation in the cortical space.

The study is well-designed with sophisticated and comprehensive data analysis. The results are robust and show strong support for a well-known model of spatial attention, the zoom-lens model. Overall, I find the results interesting and useful for the field of visual attention research.

Comments on revisions:

The authors have addressed my previous comments satisfactorily. I would encourage the authors to make data and code publicly available, which appears to be the custom in this era.

---

## [Referee Report · Reviewer #2 (Public review)]

Summary:

The study in question utilizes functional magnetic resonance imaging (fMRI) to dynamically estimate the locus and extent of covert spatial attention from visuocortical activity. The authors aim to address an important gap in our understanding of how the size of the attentional field is represented within the visual cortex. They present a novel paradigm that allows for the estimation of the spatial tuning of the attentional field and demonstrate the ability to reliably recover both the location and width of the attentional field based on BOLD responses.

Strengths:

(1) Innovative Paradigm: The development of a new approach to estimate the spatial tuning of the attentional field is a significant strength of this study. It provides a fresh perspective on how spatial attention modulates visual perception.

(2) Refined fMRI Analysis: The use of fMRI to track the spatial tuning of the attentional field across different visual regions is methodologically rigorous and provides valuable insights into the neural mechanisms underlying attentional modulation.

(3) Clear Presentation: The manuscript is well-organized, and the results are presented clearly, which aids in the reader's comprehension of the complex data and analyses involved.

Weaknesses:

(1) Lack of Neutral Cue Condition: The study does not include a neutral cue condition where the cue width spans 360{degree sign}, which could serve as a valuable baseline for assessing the BOLD response enhancements and diminishments in both attended and non-attended areas.

(2) Clarity on Task Difficulty Ratios: The explicit reasoning for the chosen letter-to-number ratios for various cue widths is not detailed. Ensuring clarity on these ratios is crucial, as it affects the task difficulty and the comparability of behavioral performance across different cue widths. It is essential that observed differences in behavior and BOLD signals are attributable solely to changes in cue width and not confounded by variations in task difficulty.

Comments on revisions:

(1) Please standardize the naming of error metrics across Figures 4-6 to improve clarity (e.g., "angular error" (Figure 4), "|angular error|" (Figure 5), and "absolute error" (Figure 6) appear to refer to the same measure). This inconsistency is also present in the main text.

(2) Consider briefly mentioning the baseline offset in Lines 179-186. It is included in Figures 4-7 and serves as a reference for interpreting attentional modulation alongside gain. Introducing it with other model parameters would improve clarity.

(3) It may be valuable to examine BOLD responses in unattended visual regions. As shown in Figure 2a, suppression patterns (e.g., the most negative responses) appear to vary in extent and distribution with attentional cue width. Analyzing these unattended regions may offer a more complete view of how attention shapes the spatial profile of cortical activity.

---

## [Author Response]

The following is the authors’ response to the original reviews

We thank the three reviewers for their insightful feedback. We look forward to addressing the raised concerns in a revised version of the manuscript. There were a few common themes among the reviews that we will briefly touch upon now, and we will provide more details in the revised manuscript.

First, the reviewers asked for the reasoning behind the task ratios we implemented for the different attentional width conditions. The different ratios were selected to be as similar as possible given the size and spacing of our stimuli (aside from the narrowest cue width of one bin, the ratios for the others were 0.66, .6 and .66). As Figure 1b shows, while the ratios were similar, task difficulty is not constant across cue widths: spreading attention makes the task more difficult generally. But, while the modeled width of the spatial distribution of attention changes monotonically with cue width, task difficulty does not. Furthermore, prior work has indicated that there is a relationship between task difficulty and the overall magnitude of the BOLD response, however we don’t suspect that this will influence the width of the modulation. How task difficulty influences the BOLD response is an important topic, and we hope that future work will investigate this relationship more directly.

Second, reviewers raised interest in the distribution of spatial attention in higher visual areas. In our study we focus only on early visual regions (V1-V3). This was primarily driven by pragmatic considerations, in that we only have retinotopic estimates for our participants in these early visual areas. Our modeling approach is dependent on having access to the population receptive field estimates for all voxels, and while the main experiment was scanned using whole brain coverage, retinotopy was measured in a separate session using a field of view only covering the occipital cortex.

Lastly, we appreciate the opportunity to clarify the purpose of the temporal interval analysis. The reviewer is correct in assuming we set out to test how much data is needed to recover the cortical modulation and how dynamic a signal the method can capture. This analysis does show that more data provides more reliable estimates, though the model was still able to recover the location and width of the attentional cue at shorter timescales of as few as two TRs. This has implications for future studies that may involve more dynamic tracking of the attentional field.

**Public Reviews**

**Reviewer #1 (Public review):**
The authors conducted an fMRI study to investigate the neural effects of sustaining attention to areas of different sizes. Participants were instructed to attend to alphanumeric characters arranged in a circular array. The size of attention field was manipulated in four levels, ranging from small (18 deg) to large (162 deg). They used a model-based method to visualize attentional modulation in early visual cortex V1 to V3, and found spatially congruent modulations of the BOLD response, i.e., as the attended area increased in size, the neural modulation also increased in size in the visual cortex. They suggest that this result is a neural manifestation of the zoomlens model of attention and that the model-based method can effectively reconstruct the neural modulation in the cortical space.The study is well-designed with sophisticated and comprehensive data analysis. The results are robust and show strong support for a well-known model of spatial attention, the zoom-lens model. Overall, I find the results interesting and useful for the field of visual attention research. I have questions about some aspects of the results and analysis as well as the bigger picture.(1) It appears that the modulation in V1 is weaker than V2 and V3 (Fig 2). In particular, the width modulation in V1 is not statistically significant (Fig 5). This result seems a bit unexpected. Given the known RF properties of neurons in these areas, in particular, smaller RF in V1, one might expect more spatially sensitive modulation in V1 than V2/V3. Some explanations and discussions would be helpful. Relatedly, one would also naturally wonder if this method can be applied to other extrastriate visual areas such as V4 and what the results look like.

We agree with the reviewer. It’s very interesting how the spatial resolution within different visual regions contributes to the overall modulation of the attentional field, and how this in turn would influence perception. Our data showed that fits in V1 appeared to be less precise than in V2 and V3. This can be seen in the goodness of fit of the model as well as the gain and absolute angular error estimates. The goodness of fit and gain were lowest in V1 and the absolute angular error was largest in V1 (see Figure 5). We speculate that the finer spatial granularity of V1 RFs was countered by a lower amplitude and SNR of attention-related modulation in V1, resulting in overall lower sensitivity to variation in attentional field width. Prior findings concur that the magnitude of covert spatial attention increases when moving from striate to extrastriate cortex (Bressler & Silver (2010); Buracas & Boynton (2007)). Notably, in our perception condition, V1 showed more spatially sensitive modulation (see Figure 7), consistent with the known RF properties of V1 neurons.

Regarding the second point: unfortunately, our dataset did not allow us to explore higherorder cortical regions with the model-based approach. While the main experiment was scanned using a sequence with whole brain coverage, the pRF estimates came from a separate scanning session which only had limited occipital coverage. Our modeling approach is dependent on the polar angle estimates from this pRF session. We now explicitly state this limitation in the methods (lines 87-89):

“In this session, the field of view was restricted to the occipital cortex to maximize SNR, thereby limiting the brain regions for which we had pRF estimates to V1, V2, and V3.”

(2) I'm a bit confused about the angular error result. Fig 4 shows that the mean angular error is close to zero, but Fig 5 reports these values to be about 30-40 deg. Why the big discrepancy? Is it due to the latter reporting absolute errors? It seems reporting the overall bias is more useful than absolute value.

The reviewer’s inference here is exactly right: Figure 4 shows signed error, whereas Figure 5 shows absolute error. We show the signed error for the example participant because, (1) by presenting the full distribution of model estimates for one participant, readers have access to a more direct representation of the data, and (2) at the individual level it is possible to examine potential directional biases in the location estimates (which do not appear to be present). As we don’t suspect a consistent directional bias across the group, we believe the absolute error in location estimates is more informative in depicting the precision in location estimates using the model-based approach. In the revised manuscript, we modified Figure 5 to make the example participant’s data visually distinct for easy comparison. We have clarified this reasoning in the text (results lines 59-64):

“The angular error distribution across blocks, separated by width condition, is shown in Figure 4 for one example participant to display block-to-block variation. The model reliably captured the location of the attentional field with low angular error and with no systematic directional bias. This result was observed across participants. We next examined the absolute angular error to assess the overall accuracy of our estimates.”

(3) A significant effect is reported for amplitude in V3 (line 78), but the graph in Fig 5 shows hardly any difference. Please confirm the finding and also explain the directionality of the effect if there is indeed one.

We realize that the y-axis scale of Figure 5 was making it difficult to see that gain decreases with cue width in area V3. Instead of keeping the y-axis limits the same across visual regions, we now adapt the y-axis scale of each subplot to the range of data values:

We now also add the direction of the effect in the text (results lines 83-86):

“We observed no significant relationship between gain and cue width in V1 and V2 (V1 t(7)=.54, p=.605; V2 t(7)=-2.19, p=.065), though we did find a significant effect in V3 illustrating that gain decreases with cue width (t(7)=-3.12, p=.017).”

(4) The purpose of the temporal interval analysis is rather unclear. I assume it has to do with how much data is needed to recover the cortical modulation and hence how dynamic a signal the method can capture. While the results make sense (i.e., more data is better), there is no obvious conclusion and/or interpretation of its meaning.

We apologize for not making our reasoning clear. We now emphasize our reasoning in the revised manuscript (results lines 110-112). Our objective was to quantify how much data was needed to recover the dynamic signal. As expected, we found that including more data reduces noise (averaging helps), but importantly, we found that we still obtained meaningful model fits even with limited data. We believe this has important implications for future paradigms that explore more dynamic deployment of spatial attention, where one would not want to average over multiple repetitions of a condition.

The first paragraph of the Temporal Interval Analysis section in the results now reads:

“In the previous analyses, we leveraged the fact that the attentional cue remained constant for 5-trial blocks (spatial profiles were computed by averaging BOLD measurements across a block of 10 TRs). We next examined the degree to which we were able to recover the attentional field on a moment-by-moment (TR-by-TR) basis. To do this, we systematically adjusted the number of TRs that contributed to the averaged spatial response profile. To maintain a constant number of observations across the temporal interval conditions, we randomly sampled a subset of TRs from each block. This allowed us to determine the amount of data needed to recover the attentional field, with a goal of examining the usability of our modeling approach in future paradigms involving more dynamic deployment of spatial attention.”

(5) I think it would be useful for the authors to make a more explicit connection to previous studies in this literature. In particular, two studies seem particularly relevant. First, how do the present results relate to those in Muller et al (2003, reference 37), which also found a zoom-lens type of neural effects. Second, how does the present method compare with spatial encoding model in Sprague & Serences (2013, reference 56), which also reconstructs the neural modulation of spatial attention. More discussions of these studies will help put the current study in the larger context.

We now make a more explicit connection to prior work in the discussion section (lines 34-54).

“We introduced a novel modeling approach that recovered the location and the size of the attentional field. Our data show that the estimated spatial spread of attentional modulation (as indicated by the recovered FWHM) consistently broadened with the cue width, replicating prior work (Müller et al., 2003; Herrmann et al., 2010). Our results go beyond prior work by linking the spatial profiles to pRF estimates, allowing us to quantify the spread of both attentional and perceptual modulation in degrees of polar angle. Interestingly, the FWHM estimates for the attentional and perceptual spatial profiles were highly similar. Additionally, for area V3 we replicate that the population response magnitude decreased with cue width (Müller et al., 2003; Feldmann-Wüstefeld and Awh, 2020). One innovation of our method is that it directly reconstructs attention-driven modulations of responses in visual cortex, setting it apart from other methods, such as inverted encoding models (e.g. Sprague & Serences, 2013). Finally, we demonstrated that our method has potential to be used in more dynamic settings, in which changes in the attentional field need to be tracked on a shorter timescale.”

(6) Fig 4b, referenced on line 123, does not exist.

We have corrected the text to reference the appropriate figure (Figure 5, results line 136).

**Reviewer #2 (Public review):**
Summary:The study in question utilizes functional magnetic resonance imaging (fMRI) to dynamically estimate the locus and extent of covert spatial attention from visuocortical activity. The authors aim to address an important gap in our understanding of how the size of the attentional field is represented within the visual cortex. They present a novel paradigm that allows for the estimation of the spatial tuning of the attentional field and demonstrate the ability to reliably recover both the location and width of the attentional field based on BOLD responses.Strengths:(1) Innovative Paradigm: The development of a new approach to estimate the spatial tuning of the attentional field is a significant strength of this study. It provides a fresh perspective on how spatial attention modulates visual perception.(2) Refined fMRI Analysis: The use of fMRI to track the spatial tuning of the attentional field across different visual regions is methodologically rigorous and provides valuable insights into the neural mechanisms underlying attentional modulation.(3) Clear Presentation: The manuscript is well-organized, and the results are presented clearly, which aids in the reader's comprehension of the complex data and analyses involved.

We thank the reviewer for summarizing the strengths in our work.

Weaknesses:(1) Lack of Neutral Cue Condition: The study does not include a neutral cue condition where the cue width spans 360°, which could serve as a valuable baseline for assessing the BOLD response enhancements and diminishments in both attended and non-attended areas.

We do not think that the lack of a neutral cue condition substantially limits our ability to address the core questions of interest in the present work. We set out to estimate the locus and the spread of covert spatial attention. By definition, a neutral cue does not have a focus of attention as the whole annulus becomes task relevant. We agree with the reviewer that how spatial attention influences the magnitude of the BOLD response is still not well defined; i.e., does attending a location multiplicatively enhance responses at an attended location or does it instead act to suppress responses outside the focus of attention? A neutral cue condition would be necessary to be able to explore these types of questions. However, our findings don’t rest on any assumptions about this. Instead, we quantify the attentional modulation with a model-based approach and show that we can reliably recover its locus, and reveal a broadening in the attentional modulation with wider cues.

We realize that throughout the original manuscript we often used the term ‘attentional enhancement,’ which might inadvertently specify an increase with respect to a neutral condition. To be more agnostic to the directionality of the effect, we have changed this to ‘attentional modulation’ and ‘attentional gain’ throughout the manuscript. Additionally, we have added results and visualizations for the baseline parameter to all results figures (Figures 4-7) to help readers further interpret our findings.

(2) Clarity on Task Difficulty Ratios: The explicit reasoning for the chosen letter-to-number ratios for various cue widths is not detailed. Ensuring clarity on these ratios is crucial, as it affects the task difficulty and the comparability of behavioral performance across different cue widths. It is essential that observed differences in behavior and BOLD signals are attributable solely to changes in cue width and not confounded by variations in task difficulty.

The ratios were selected to be as similar as possible given the size and spacing of our stimuli (aside from the narrowest cue width of one bin, the proportions for the others were 0.67, 0.60, and 0.67). We have updated the methods section to state this explicitly (methods lines 36-38):

“The ratios were selected to be as similar as possible given the size and spacing of our stimuli (aside from the one-bin cue, the proportions for the other cues were 0.67, 0.60, 0.67).”

As Figure 1b shows, task accuracy showed small and non-monotonic changes across the three larger cue widths, dissociable from the monotonic pattern seen for the modelestimated width of the attentional field. Furthermore, as prior work has indicated that there is a relationship between task difficulty and the overall magnitude of the BOLD response (e.g., Ress, Backus & Heeger, 2000), we would primarily expect effects of task difficulty on the gain or baseline rather than the width. How exactly task difficulty influences the BOLD response and whether this would, in fact, interact with the width of the attentional field is an important topic, and we hope that future work will investigate this relationship more directly.

We have clarified these points within the text, and now explicitly motivate future work looking at these important interactions (discussion lines 57-67):

“The observed effects of attentional field width were unlikely to be directly attributable to variation in task difficulty. Participants' task in our study was to discriminate whether more numbers or more letters were presented within a cued region of an iso-eccentric annulus of white noise. For our different cue widths, the ratios of numbers and letters were selected to be as similar as possible given the size and spacing of our stimuli. Changes in accuracy across the three larger cue widths were small and non-monotonic, implying task difficulty was dissociable from width per se. This dissociation bolsters the interpretability of our model fits; nevertheless, future work should further investigate how task difficulty interacts with the spread of the attentional field and the amplitude of attention-related BOLD effects (cf. Ress, Backus & Heeger, 2000).”

**Reviewer #3 (Public review):**
Summary:In this report, the authors tested how manipulating the contiguous set of stimuli on the screen that should be used to guide behavior - that is, the scope of visual spatial attention - impacts the magnitude and profile of well-established attentional enhancements in visual retinotopic cortex. During fMRI scanning, participants attended to a cued section of the screen for blocks of trials and performed a letter vs digit discrimination task at each attended location (and judged whether the majority of characters were letters/digits). Importantly, the visual stimulus was identical across attention conditions, so any observed response modulations are due to topdown task demands rather than visual input. The authors employ population receptive field (pRF) models, which are used to sort voxel activation with respect to the location and scope of spatial attention and fit a Gaussian-like function to the profile of attentional enhancement from each region and condition. The authors find that attending to a broader region of space expands the profile of attentional enhancement across the cortex (with a larger effect in higher visual areas), but does not strongly impact the magnitude of this enhancement, such that each attended stimulus is enhanced to a similar degree. Interestingly, these modulations, overall, mimic changes in response properties caused by changes to the stimulus itself (increase in contrast matching the attended location in the primary experiment). The finding that attentional enhancement primarily broadens, but does not substantially weaken in most regions, is an important addition to our understanding of the impact of distributed attention on neural responses, and will provide meaningful constraints to neural models of attentional enhancement.Strengths:(1) Well-designed manipulations (changing location and scope of spatial attention), and careful retinotopic/pRF mapping, allow for a robust assay of the spatial profile of attentional enhancement, which has not been carefully measured in previous studies.(2) Results are overall clear, especially concerning width of the spatial region of attentional enhancement, and lack of clear and consistent evidence for reduction in the amplitude of enhancement profile.(3) Model-fitting to characterize spatial scope of enhancement improves interpretability of findings.

We thank the reviewer for highlighting the strengths of our study.

Weaknesses:(1) Task difficulty seems to vary as a function of spatial scope of attention, with varying ratios of letters/digits across spatial scope conditions, which may complicate interpretations of neural modulation results

The reviewer is correct in observing that task accuracy varied across cue widths. Though we selected the task ratios to be as similar as possible given the size and spacing of our stimuli (aside from the narrowest cue width of one bin, the proportions for the others were 0.67, 0.60, and 0.67), behavioral accuracy across the three larger cue widths was not identical. Prior research has shown that there is a relationship between task difficulty and the overall magnitude of the BOLD response (e.g., Ress, Backus & Heeger, 2000). Thus, we would primarily expect effects of task difficulty on gain rather than width. How task difficulty influences the BOLD response and whether this would, in fact, interact with the width of the attentional field is an important topic, and we hope that future work will investigate this relationship more directly.

To clarify these points and highlight the potential for future work looking at these important interactions, we added the following text to the discussion section (discussion lines 57-67):

“The observed effects of attentional field width were unlikely to be directly attributable to variation in task difficulty. Participants' task in our study was to discriminate whether more numbers or more letters were presented within a cued region of an iso-eccentric annulus of white noise. For our different cue widths, the ratios of numbers and letters were selected to be as similar as possible given the size and spacing of our stimuli. Changes in accuracy across the three larger cue widths were small and non-monotonic, implying task difficulty was dissociable from width per se. This dissociation bolsters the interpretability of our model fits; nevertheless, future work should further investigate how task difficulty interacts with the spread of the attentional field and the amplitude of attention-related BOLD effects (cf. Ress, Backus and Heeger, 2000).”

(2) Some aspects of analysis/data sorting are unclear (e.g., how are voxels selected for analyses?)

We apologize for not describing our voxel selection in sufficient detail. Some of the questions raised in the private comments are closely related to this point, we therefore aim to clarify all concerns below:

- Voxel selection: To select voxels that contribute to the 1D spatial profiles, we relied on the independent pRF dataset. We first defined some general requirements that needed to be met. Specifically, (1) the goodness of fit (R^2^) of the pRF fits needed to be greater than 10%; (2) the estimated eccentricity had to fall within [0.7 9.1] degree eccentricity (to exclude voxels in the fovea and voxels with estimated eccentricities larger than the pRF mapping stimulus); (3) the estimated size must be greater than 0.01 degree visual angle.

Next, we included only voxels whose pRF overlapped with the white noise annulus. Estimated eccentricity was used to select all voxels whose eccentricity estimate fell within the annulus bounds. However, here it is also important to take the size of the pRF into account. Some voxels’ estimated eccentricity might fall just outside the annulus, but will still have substantial overlap due to the size of their pRF. Therefore, we further included all voxels whose estimated pRF size resulted in overlap with the annulus.

This implies that some voxels with greater eccentricities and larger pRF sizes contribute to the 1D profile, which will influence the spatial specificity of the 1D profiles. However, we want to emphasize that in our view, the exact FWHM value is not so much of interest, as this will always be dependent on the voxel selection and many other data processing steps. Instead, we focus on the relative differences of the FWHM driven by the parametric attentional cue width manipulation.

- Data sorting and binning. The reviewer raises an important point about how the FWHM value should be interpreted considering the data processing steps. To generate the 1D spatial profile, we binned voxels based on their estimated polar angle preference into 6degree bins and applied a moving average of 18 degrees to smooth the 1D profiles. Both of these processing steps will influence the spatial specificity of the profile. The binning step facilitates recentering based on cue center and combining across trials.

To explore the extent to which the moving average substantially impacted our results, we reran our analyses without that smoothing step. The vast majority of the results held. In V1, we found a significant effect of cue width on FWHM where the result was not significant previously (*t*(7)=2.52, *p*=.040). Additionally, when looking at the minimum number of TRs needed to see a significant effect of cue width on FWHM, without the smoothing step in V1 it took 10 TRs (not significant at 10 TRs previously), in V2 it took 5 TRs (10 previously), and in V3 it took 3 TRs (2 previously). The other notable difference is that FWHM was generally a bit larger when the moving average smoothing was performed. We have visualized the group results for the FWHM estimates below to help with comparison.

**Author response image 1. sa3fig1:** No moving average smoothing.

Voxel selection methods have been clarified in methods section lines 132-139:

“Within each ROI, pRF modeling results were used to constrain voxel selection used in the main experiment. We excluded voxels with a preferred eccentricity outside the bounds of the pRF stimulus (<0.7° and >9.1°), with a pRF size smaller than 0.01°, or with poor spatial selectivity as indicated by the pRF model fit (R2 < 10%). Following our 2D visualizations (see below), we further constrained voxel selection by only including voxels whose pRF overlapped with the white noise annulus. We included all voxels with an estimated eccentricity within the annulus bounds, as well as voxels with an estimated pRF size that would overlap the annulus.”

Data binning methods have been clarified in methods section lines 154-159:

“Voxels with pRFs overlapping the white noise annulus were grouped into 60 bins according to their pRF polar angle estimate (6° polar angle bin width). We computed a median BOLD response within each bin. This facilitated the recentering of each profile to align all cue centers for subsequent combining across trials. To improve the signal-to-noise ratio, the resulting profile was smoothed with a moving average filter (width 18° polar angle; see Figure 2b).”

(3) While the focus of this report is on modulations of visual cortex responses due to attention, the lack of inclusion of results from other retinotopic areas (e.g. V3AB, hV4, IPS regions like IPS0/1) is a weakness

We agree with the reviewer that using this approach in other retinotopic areas would be of significant interest. In this case, population receptive field mapping occurred in a separate session with a field of view only covering the occipital cortex (in contrast to the experimental session, which had whole-brain coverage). Because our modeling approach relies on these pRF estimates, we were unable to explore higher visual areas. However, we hope future work will follow up on this.

We have added the following text to the methods section describing the pRF mapping session (lines 87-89):

“In this session, the field of view was restricted to the occipital cortex to maximize SNR, thereby limiting the brain regions for which we had pRF estimates to V1, V2, and V3.”

(4) Additional analyses comparing model fits across amounts of data analyzed suggest the model fitting procedure is biased, with some parameters (e.g., FWHM, error, gain) scaling with noise.

In this analysis, we sought to test how much data was needed to recover the attentional field, in view of the need for additional fMRI-based tools for use in tasks that involve more rapid dynamic adaptation of attention. Though we did find that more data reduced noise (and accordingly decreased absolute error and amplitude while increasing FWHM and R^2^), absolute angular error remained low across different temporal intervals (well below the chance level of 90°). With regard to FWHM, we believe that the more important finding is that the model-estimated FWHM was modulated by cue width at shorter timescales of as few as two TRs while maintaining relatively low angular error. We refrain from drawing conclusions here on the basis of the exact FWHM values, both because we don’t have a ground truth for the attentional field and because various processing pipeline steps can impact the values as well. Rather, we are looking at relative value and overall patterns in the estimates. The observed patterns imply that the model recovers meaningful modulation of the attentional field even at shorter time scales.

**Recommendations for the authors:**

**Reviewer #1 (Recommendations for the authors):**
Additional data reporting and discussion of results are needed as outlined in the public review.
**Reviewer #2 (Recommendations for the authors):**
(1) The current experimental design effectively captured the impact of varying cue widths on the BOLD response in the visual cortex. However, the inclusion of a neutral cue condition, where the cue width spans 360{degree sign} and all peripheral stimuli are attended, could serve as a valuable baseline. This would enable a quantitative assessment of how much the BOLD response is enhanced in specific spatial regions due to focused cues and, conversely, how much it is diminished in non-attended areas, along with the spatial extent of these effects.

Please refer to our response in the public review.

(2) While the study provides valuable insights into BOLD signal changes in visual areas corresponding to the focus of attention, it does not extend its analysis to the impact on regions outside the focus of attention. It would be beneficial to explore whether there is a corresponding decrease in BOLD signal in non-attended regions, and if identified, to describe the spatial extent and position of this effect relative to the attended area. Such an analysis could yield deeper insights into how attention influences activity across the visual cortex.

We agree with the reviewer that it is very interesting to examine the spread of attention across the whole visual field. Our experiment was designed to focus on width modulations at a fixed eccentricity, but future work should explore how the attentional field changes with eccentricity and interacts with spatial variations across the visual field. This is highlighted in our discussion section (lines 76-81):

“Future work can help provide a better understanding of the contribution of spatial attention by considering how the attentional field interacts with these well described spatial variations across the visual field. Measuring the full spatial distribution of the attentional field (across both eccentricity and polar angle) will shed light on how spatial attention guides perception by interacting with the non-uniformity of spatial representations.”

The addition of figure panels for the estimated baseline parameter in Figures 4-7 provides further information about BOLD effects in unattended regions of the annulus.

(3) The rationale behind the selection of task difficulty ratios for different cue widths, specifically the letter-to-number ratios of 1:0, 1:2, 2:3, and 3:6 (or vice versa) for cue widths of 18{degree sign}, 54{degree sign}, 90{degree sign}, and 162{degree sign} respectively, was not explicitly discussed. It would be beneficial to clarify the basis for these ratios, as they may influence the perceived difficulty of the task and thus the comparability of behavioral performance across different cue widths. Ensuring that the task difficulty is consistent across conditions is crucial for attributing differences in behavior and BOLD signals solely to changes in cue width and not confounded by variations in task difficulty.

Please refer to our response in the public review. We now clarify why we selected these ratios, and acknowledge more explicitly that behavioral performance differed across width conditions. See also our reply to private comment 1 from Reviewer 3 for some additional analyses examining task related influences.

**Reviewer #3 (Recommendations for the authors):**
(1) Task difficulty: the task seems exceptionally challenging. Stimuli are presented at a relativelyeccentric position for a very brief duration, and a large number of comparisons must be made across a broad region of space. This is reflected in the behavioral performance, which decreases rapidly as the scope of attention increases (Fig. 1). Because trials are blocked, does this change in task difficulty across conditions impact the degree to which neural responses are modulated? How should we consider differences in task difficulty in interpreting the conclusions (especially with respect to the amplitude parameter)? Also, note that the difficulty scales both with number of stimuli - as more need to be compared - but also with the ratio, which differs nonmonotonically across task conditions. One way to dissociate these might be RT: for 54/162, which both employ the same ratio of letter/digits and have similar accuracy, is RT longer for 162, which requires attending more stimuli?

In addition to our comments in response to the public review, we emphasize that the reviewer makes an important point that there are differences in task difficulty, though the ratios are as close as they can be given the size and spacing of our stimuli. Behavioral performance varied non-monotonically with cue width, bolstering our confidence that our monotonically increasing model-estimated width is likely not entirely driven by task difficulty. There nevertheless remain open questions related to how task difficulty does impact BOLD attentional modulation, which we hope future work will more directly investigate.

The reviewer's comments identify two ways our data might preliminarily speak to questions about BOLD attentional modulation and task difficulty. First: how might the amplitude parameter reflect task difficulty? This is an apt question as we agree with the reviewer that it would be a likely candidate in which to observe effects of task difficulty. We do find a small effect of *cue width* on our amplitude estimates (amplitude decreases with width) in V3. Using the same analysis technique to look at the relationship between task difficulty and amplitude, we find no clear relationship in any of the visual areas (all *p* >= 0.165, testing whether the slopes differed from zero at the group level using a one-sample t-test). We believe future work using other experimental manipulations should look more systematically at the relationship between task difficulty and amplitude of the attentional BOLD enhancement.

Second: Does the same ratio at different widths elicit different behavioral responses (namely accuracy and RT)? We followed the reviewer’s suggestion to compare performance between cue widths of three and nine (identical ratios, different widths; see Author response image 2 and Figure 5). We found that, using a paired t-test, behavioral accuracy differed between the two cue widths (mean accuracy of 0.73 versus 0.69, p = 0.008), with better performance for cue width three. RT did not differ significantly between the two conditions (paired t-test, p = 0.729). This could be due to the fact that participants were not incentivized to respond as quickly as possible, they merely needed to respond before the end of the response window (1.25 s) following the stimulus presentation (0.5 s). The comparisons for accuracy and RT (calculated from time of stimulus appearance) are plotted below:

**Author response image 2. sa3fig2:** 

In summary, with matched stimulus ratios, the wider cue was associated with worse (though not slower) performance. This could be due to the fact that more elements are involved and/or that tasks become more difficult when attending to a broader swath of space. Given these results, we believe that future studies targeting difficulty effects should use direct and independent manipulations of task difficulty and attentional width.

(2) Eye movements: while the authors do a good job addressing the average eccentricity of fixation, I'm not sure this fully addresses concerns with eye movements, especially for the character-discrimination task which surely benefits from foveation (and requires a great deal of control to minimize saccades!). Can the authors additionally provide data on, e.g., # of fixations within the attended stimulus annulus, or fixation heatmap, or # of saccades, or some other indicator of likelihood of fixating the letter stimuli for each condition?

We agree with the reviewer that this task is surely much easier if one foveated the stimuli, and it did indeed require control to minimize saccades to the annulus. (We appreciate the effort and motivation of our participants!) We are happy to provide additional data to address these reasonable concerns about eye movements. Below, we have visualized the number of fixations to the annulus, separated by participant and width. Though there is variability across participants, there are at most 16 instances of fixations to the annulus for a given participant, combined across all width conditions. The median number of fixations to the annulus per width is zero (shown in red). Considering the amount of time participants engaged in the task (between 8 and 12 runs of the task, each run with 100 trials), this indicates participants were generally successful at maintaining central fixation while the stimuli were presented.

**Author response image 3. sa3fig3:** 

We added the results of this analysis to the methods section (lines 205-208):

“Additionally, we examined the number of fixations to the white noise annulus itself. No participant had more than 16 fixations (out of 800-1200 trials) to the annulus during the task, further suggesting that participants successfully maintained fixation.”

(3) pRF sorting and smoothing: Throughout, the authors are analyzing data binned based on pRF properties with respect to the attended location ("voxels with pRFs overlapping with the white noise annulus", line 243-244) First, what does this mean? Does the pRF center need to be within the annulus? Or is there a threshold based on the pRF size? If so, how is this implemented? Additionally, considering the methods text in lines 242-247, the authors mention that they bin across 6 deg-wide bins and smooth with a moving average (18 deg), which I think will lead to further expansion of the profile of attentional enhancement (see also below)

We provide a detailed response in the public review. Furthermore, we have clarified the voxel selection procedure in the Methods (lines 132–139 & 154–159).

(4) FWHM values: The authors interpret the larger FWHMs estimated from their model-fitting than the actual size of the attended region as a meaningful result. However, depending on details of the sorting procedure above, this may just be due to the data processing itself. One way to identify how much expansion of FWHM occurs due to analysis is by simulating data given estimates of pRF properties for a 'known' shape of modulation (e.g., square wave exactly spanning the attended aperture) and compare the resulting FWHM to that observed for attention and perception conditions (e.g., Fig. 7c).

We provide a detailed response in the public review. The essence of our response is to refrain from interpreting the precise recovered FWHM values, which will be influenced by multiple processing steps, and instead to focus on relative differences as a function of the attentional cue width. Accordingly, we did not add simulations to the revised manuscript, although we agree with the reviewer that such simulations could shed light on the underlying spatial resolution, and how binning and smoothing influences the estimated FWHM. We have clarified our interpretation of FWHM results in the manuscript as follows:

Results lines 137-141:

“One possibility is that the BOLD-derived FWHM might tend to overestimate the retinotopic extent of the modulation, perhaps driven by binning and smoothing processing steps to create the 1D spatial profiles. If this were the case, we would expect to obtain similar FWHM estimates when modeling the perceptual modulations as well.”

Results lines 169-175:

“Mirroring the results from the attentional manipulation, FWHM estimates systematically exceeded the nominal size of the perceptually modulated region of the visual field. Comparing the estimated FWHMs of the perceptual and attentional spatial profiles (Figure 7c) revealed that the estimated widths were highly comparable (Pearson correlation r=0.664 across width conditions and visual regions). Importantly, the relative differences in FWHM show meaningful effects of both cue and contrast width in a similar manner for both attentional and perceptual forms of modulation.”

Discussion lines 16-22:

“We also found that the estimated spatial spread of the attentional modulation (as indicated by the recovered FWHM) was consistently wider than the cued region itself. We therefore compared the spread of the attention field with the spatial profile of a perceptually induced width manipulation. The results were comparable in both the attentional and perceptual versions of the task, suggesting that cueing attention to a region results in a similar 1D spatial profile to when the stimulus contrast is simply increased in that region.”

(5) Baseline parameter: looking at the 'raw' response profiles shown in Fig. 2b, it looks, at first, like the wider attentional window shows substantially lower enhancement. However, this seems to be mitigated by the shift of the curve downwards. Can the authors analyze the baseline parameter in a similar manner as their amplitude analyses throughout? This is especially interesting in contrast to the perception results (Fig. 7), for which the baseline does not seem to scale in a similar way.

We agree with the reviewer that the baseline parameter is worth examining, and have therefore added panels displaying the baseline parameter into all results figures (Figures 4-7). There was no significant association between cue width and baseline offset in any of the three visual regions.

(6) Outlier: Fig. 5, V2, Amplitude result seems to have a substantial outlier - is there any notable difference in e.g. retinotopy in this participant?

One participant indeed has a notably larger median amplitude estimate in V2. Below, we plot the spatial coverage from the pRF data for this participant (022), as well as all other participants.

**Author response image 4. sa3fig4:** 

Each subplot represents a participant's 2D histogram of included voxels for the 1D spatial profiles; the colors indicate the proportion of voxels that fell within a specific x,y coordinate bin. Note that this visualization only shows x and y estimates and does not take into account size of the pRF. While there is variation across participants in the visual field coverage, the overall similarity of the maps indicates that retinotopy is unlikely to be the explanation.

To further explore whether this participant might be an outlier, we additionally looked at behavioral performance, angular error and FWHM parameters as well as the goodness of fit of the model. On all these criteria this participant did not appear to be an outlier. We therefore see no reason to exclude this participant from the analyses.

(7) Fig. 4 vs Fig. 5: I understand that Fig. 4 shows results from a single participant, showing variability across blocks, while Fig. 5 shows aggregate results across participants. However, the Angular Error figure shows complementary results - Fig. 4 shows the variability of best-fit angular error, while Fig. 5 shows the average deviation (approximately the width of the error distribution). This makes sense I think, but perhaps the abs(error) for the single participant shown in Fig. 4 should be included in the caption so we can easily compare between figures.

That's right: the Figure 4 results show the signed error, whereas the Figure 5 results show the absolute error. We agree that reporting the absolute error values for the example participant would facilitate comparison. Rather than add the values to the text, we have made the example participant’s data visually distinct within Figure 5 for easy comparison.

(8) Bias in model fits: the analysis shown in Fig. 6 compares the estimated parameters across amounts of data used to compute attentional modulation profiles for fitting those parameters. If the model-fitting procedure were unbiased, my sense is we would likely see no impact of the number of TRs on the parameters (R^2 should improve, abs(error) should improve, but FWHM, amplitude, baseline, etc should be approximately stable, if noisier). However, instead, it looks like more/less data leads to biased estimates, such that FWHM is biased to be smaller with more noise, and amplitude is biased to be larger. This suggests (to me) that the fit is landing on a spiky function that captures a noise wiggle in the profile. I don't think this is a problem for the primary results across the whole block of 10 TRs, which is the main point of the paper. Indeed, I'm not sure what this figure is really adding, since the single-TR result isn't pursued further (see below).

Please refer to our response in the public review, comment 4.

(9) 'Dynamics': The paper, starting in the title, claims to get at the 'dynamics' of attention fields. At least to me, that word implies something that changes over time (rather than across trials). Maybe I'm misinterpreting the intent of the authors, but at present, I'm not sure the use of the word is justified. That said, if the authors could analyze the temporal evolution of the attention field through each block of trials at 1- or 2-TR resolution, I think that could be a neat addition to the paper and would support the claim that the study assays dynamic attention fields.

We thank the reviewer for giving us a chance to speak more directly to the dynamic aspect of our approach. Here, we specifically use the word “dynamic” to refer to trial-to-trial dynamics. Importantly, our temporal interval analysis suggests that we can recover information about the attentional field at a relatively fine-grained temporal resolution (a few seconds, or 2 TRs). Following this methodological proof-of-concept to dynamically track the attentional field, we are excited about future work that can more directly investigate the manner in which the attentional field evolves through time, especially in comparison to other methods that first require training on large amounts of data.

(10) Correction for multiple comparisons across ROIs: it seems that it may be necessary to correct statistical tests for multiple comparisons across each ROI (e.g., Fig. 5 regression tests). If this isn't necessary, the authors should include some justification. I'm not sure this changes any conclusions, but is worth considering.

We appreciate the opportunity to explain our reasoning regarding multiple comparisons. We thought it appropriate not to correct as we are not comparing across regions and are not treating tests of V1, V2, and V3 as multiple opportunities to support a common hypothesis. Rather, the presence or absence of an effect in each visual region is a separate question. We would typically perform correction for multiple comparisons to control the familywise error rate when conducting a family of tests addressing a common hypothesis. We have added this to the Methods section (lines 192-195):

“No multiple comparison correction was applied, as the different tests for each region are treated as separate questions. However, using a threshold of 0.017 for p-values would correct for comparisons across the three brain regions.”

However, we are happy to provide corrected results. If we use Bonferroni correction across ROIs (i.e. multiply p-values by three), there are some small changes from significant to only trending towards significance, but these changes don’t affect any core results. The changes that go from significant to trending are:

Associated with Figure 5 – In V3, the relationship of cue width to amplitude goes from a p-value of 0.017 to 0.051.

Associated with Figure 6 –

V1: the effect of cue width on FWHM goes from p = 0.043 to 0.128.

V2: the effect of TR on both FWHM and R2 goes from p = ~0.02 to ~0.06.

V3: the effect of cue width on amplitude goes from p = 0.024 to 0.073.